# ChaosBench: A Multi-Channel, Physics-Based Benchmark for Subseasonal-to-Seasonal Climate Prediction

**Juan Nathaniel[1,*], Yongquan Qu[1], Tung Nguyen[2], Sungduk Yu[3,5], Julius Busecke[1,4], Aditya Grover[2], Pierre Gentine[1]**
[1]Columbia University, [2]UCLA, [3]UCI, [4]LDEO, [5] Intel Labs

## Abstract

Accurate prediction of climate in the subseasonal-to-seasonal scale is crucial for disaster preparedness and robust decision making amidst climate change. Yet, forecasting beyond the weather timescale is challenging because it deals with problems other than initial condition, including boundary interaction, butterfly effect, and our inherent lack of physical understanding. At present, existing benchmarks tend to have shorter forecasting range of up-to 15 days, do not include a wide range of operational baselines, and lack physics-based constraints for explainability. Thus, we propose ChaosBench, a challenging benchmark to extend the predictability range of data-driven weather emulators to S2S timescale. First, ChaosBench is comprised of variables beyond the typical surface-atmospheric ERA5 to also include ocean, ice, and land reanalysis products that span over 45 years to allow for full Earth system emulation that respects boundary conditions. We also propose physics-based, in addition to deterministic and probabilistic metrics, to ensure a physically-consistent ensemble that accounts for butterfly effect. Furthermore, we evaluate on a diverse set of physics-based forecasts from four national weather agencies as baselines to our data-driven counterpart such as ViT/ClimaX, PanguWeather, GraphCast, and FourCastNetV2. Overall, we find methods originally developed for weather-scale applications fail on S2S task: their performance simply collapse to an unskilled climatology. Nonetheless, we outline and demonstrate several strategies that can extend the predictability range of existing weather emulators, including the use of ensembles, robust control of error propagation, and the use of physics-informed models. Our benchmark, datasets, and instructions are available at https://leap-stc.github.io/ChaosBench.

## 1 Introduction

Although critical for economic planning, disaster preparedness, and policy-making, subseasonal-to-seasonal (S2S) prediction is lagging behind the more established field of short/medium-range weather, or long-range climate predictions. For instance, many natural hazards tend to manifest in the S2S scale, including the slow-onset of droughts that lead to wildfire [1, 2], heavy precipitations that lead to flooding [3], and persistent weather anomalies that lead to extremes [4]. So far, current approaches to weather and climate prediction are heavily reliant on physics-based models in the form of Numerical Weather Prediction (NWP). Many NWPs are based on the discretization of governing equations that describe thermodynamics, fluid flows, *etc*. However, these models are expensive to run especially in high-resolution setting. For example, there are massive computational overheads to perform numerical integration at fine spatiotemporal resolutions that are operationally useful [5]. Furthermore,

---

*Corresponding author: jn2808@columbia.edu

38th Conference on Neural Information Processing Systems (NeurIPS 2024) Track on Datasets and Benchmarks.

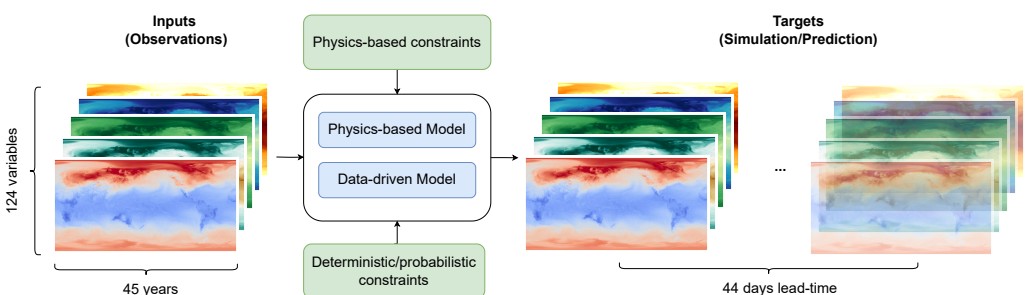

Figure 1: We propose ChaosBench, a large-scale, fully-coupled, physics-based benchmark for subseasonal-to-seasonal (S2S) climate prediction. It is framed as a high-dimensional sequential regression task that consists of 45+ years, multi-system observations for validating physics-based and data-driven models, and training the latter. Physics-based forecasts are generated from four national weather agencies with 44-day lead-time and serve as baselines to data-driven forecasts. Our benchmark is one of the first to incorporate physics-based metrics to ensure physically-consistent and explainable models. The blurred image at $\Delta t = 44$ represents a challenge of long-term forecasting.

their relative inaccessibility to non-experts is a major roadblock to the broader community. As a result, there is a growing interest to apply data-driven models to emulate NWPs, as they tend to have faster inference speed, are less resource-hungry, and more accessible [6, 7, 8, 9, 10, 11, 12]. Nevertheless, many data-driven benchmarks have so far been focused on the short (1-5 days), medium (5-15 days), and long (years-decades) forecasting ranges. In this work, we include S2S as a more challenging task that requires different emulation strategies: being in between two extremes, it is doubly sensitive to (1) *initial conditions* (IC) as in the case for short/medium-range weather, and (2) *boundary conditions* (BC) as in the case for long-range climate [13, 14, 15, 16].

We propose ChaosBench to bridge these gaps (Figure 1). It is comprised of variables beyond the typical surface-atmospheric ERA5 to also include ocean, ice, and land reanalysis products that span over 45 years to allow for full Earth system emulation that respects boundary processes. We also provide 44-day ahead physics-based control (deterministic) and perturbed (ensemble) forecasts from four national weather agencies over the last 8 years as baselines. In addition, we introduce physics-based and incorporate probabilistic, in addition to deterministic metrics, for a more physically-consistent ensemble that accounts for butterfly effect. As far as we know, ChaosBench is one of the first to systematically evaluate several state-of-the-art data-driven models including ViT/ClimaX [17], PanguWeather [18], GraphCast [7], and FourCastNetV2 [9] on S2S predictability.

In this work, we demonstrate that existing physics-based and data-driven models are indistinguishable from unskilled climatology as the forecasting range approaches the S2S timescale. The high spectral divergence observed in many state-of-the-art models suggests the lost of predictive accuracy of multi-scale structures. This leads to significant blurring and a tendency towards smoother predictions. For one, such averaging is of little use when one attempts to identify extreme events requiring high-fidelity forecasts on the S2S scale (e.g., regional droughts, hurricanes, *etc*). Also, performing comparably worse than climatology renders them *operationally unusable*. This highlights the urgent need for a robust and unified data-driven S2S intercomparison project.

## 2   Related Work

In recent years, several benchmarks have been introduced to push the field of data-driven weather and climate prediction [19, 20, 21, 22, 23, 24, 25, 26, 27]. We analyze the limitations of existing works, and propose how ChaosBench fills in these gaps (see Table 1, more justifications in Appendix C).

**Gap in forecast lead-time**. Many existing benchmarks are built for short/medium-range weather (up to 15 days) [22, 19, 20], and long-term climate (annual to decadal scale) [26]. As discussed earlier, these problems tend to be easier due to the lack of combined sensitivities to IC and BC [13, 14].

**Limited spatiotemporal extent**. Many S2S benchmarks tend to focus on regional forecasts, such as the US [23, 24]. In addition, the temporal extent of observation with common interval is more varied,

Table 1: Comparison with other benchmark datasets: ChaosBench (ours) is evaluated on the largest set of global variables, benchmarked against large number of operational NWPs (four national agencies in the US, Europe, UK, and Asia), and incorporates both physics-based and probabilistic metrics for a more physically-consistent S2S ensemble forecast.

| Datasets | # input variables | # target variables | forecast lead (days) | physics-based metrics | probabilistic metrics | spatial extent |
|---|---|---|---|---|---|---|
| WeatherBench [22] | 110 | 110 | 15 | ✓ | ✓ | global |
| SubseasonalRodeo [23] | <30 | 2 | 44 | ✗ | ✗ | western US |
| SubseasonalClimateUSA [24] | <30 | 2 | 44 | ✗ | ✓ | contiguous US |
| CliMetLab [25] | <30 | 2 | 44 | ✗ | ✓ | global |
| **ChaosBench (ours)** | 124 | 124 | 44 | ✓ | ✓ | global |

with some less than 20 years [19, 25]. ChaosBench has the most extensive overlapping temporal coverage yet, extending to 45+ years of inputs covering multiple reanalysis products beyond ERA5.

**Limited diversity of baseline models**. Having a large set of physics-based forecasts as baselines is key to reducing bias and diversifying the target goal-posts. Previous benchmarks are mostly focused on increasing the number of data-driven models for baselines [22, 23]. In contrast, ChaosBench also places weights on expanding the diversity of physics-based models, including those operated by leading national weather agencies in the US, Europe, UK, and Asia.

**Lack of physics-based constraints**. So far, limited number of benchmarks have explicitly incorporated physical principles to improve or constrain forecasts. ChaosBench introduces physics-based metrics that can be used for comparison (*scalar*) and integrated into ML pipeline (*differentiable*).

## 3 ChaosBench

### 3.1 Observations

We discuss the components of ChaosBench, including the global reanalysis products of surface-atmosphere (ERA5), sea-ice (ORAS5), and terrestrial (LRA5), as well as simulations from physics-based models. The spatiotemporal resolutions of the former are matched with the latter's daily forecasts at $1.5°$ to allow for consistent evaluation and integration e.g., hybrid physics-based emulator. However, we provide a one-liner script to process higher e.g., $0.25°$ resolution input in Section B.4.

**ERA5** Reanalysis provides a comprehensive record of the global atmosphere combining physics and observations for correction [28]. We processed their hourly data from 1979 to present and selected measurements at the 00UTC step. The variables include temperature ($t$), specific humidity ($q$), geopotential height ($z$), and 3D wind speed ($u, v, w$) at 10 pressure levels: $1000, 925, 850, 700, 500, 300, 200, 100, 50, 10$ hpa, totalling 60 variables (full list in D.1.1).

**ORAS5** or the Ocean Reanalysis System 5 provides an extensive record of sea-ice variables that incorporate multiple depth levels [29]. Since the public data is available on a monthly basis, we replicate them for daily compatibility with temporal extent from 1979 to present, for a total of 21 variables, including `sst` and `ssh` (full list in D.1.2).

**LRA5** or ERA5-Land Reanalysis provides a detailed record of variables governing global terrestrial processes with specific corrections tailored for land surface applications such as flood forecasting [30] or carbon fluxes [21, 31]. We processed hourly data from 1979 to present and selected measurements at the 00UTC step, for a total of 43 variables, including `t2m`, `u10`, `v10`, and `tp` (full list in D.1.3).

### 3.2 Simulations

We briefly describe the forecast generation process from physics-based models (Figure 2), including details on forecast frequency and the number of ensemble members. More details are provided in

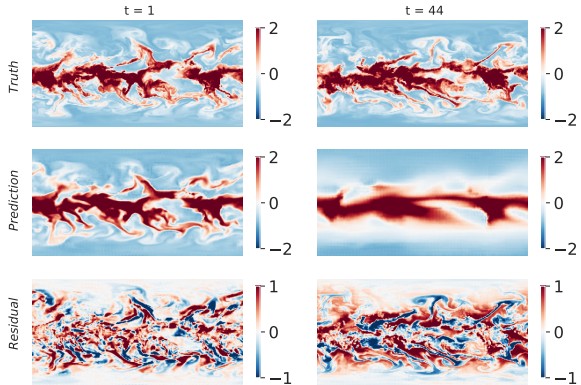

| | t = 1 | t = 44 |

(a) Normalized humidity@700-hpa label, forecast, and residual at the first ($t = 1$) and final ($t = 44$) step with ClimaX

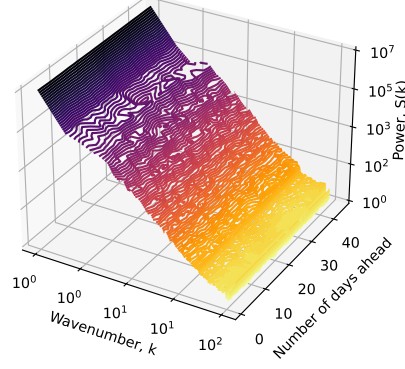

(b) Power spectrum $S(k)$ *vs.* wavenumber $k$ plot as a function of prediction step of normalized humidity@700-hpa with ClimaX

Figure 3: Motivating problem: as we perform longer rollouts, the (a) residual error becomes larger and prediction becomes blurry. This behavior is captured in the Fourier frequency domain where the (b) power spectra $S(k)$ at low wavenumber $k$ (i.e., low frequency signal) remains consistent at long rollouts, but not for higher $k$ (i.e., high frequency signal). This phenomenon explains why long-term forecasts excel at capturing *large-scale pattern* but not *fine-grained details* i.e., smooth.

Appendix D.2. The list of available variables for physics-based forecast are similar to ERA5 but missing {q10,q50,q100} and w $\notin$ {w500} for a total of 48 variables. In all, we process control (deterministic) and perturbed (ensemble) forecasts from 2016 to present [32].

**UKMO**. The UK Meteorological Office uses the Global Seasonal Forecast System Version 6 (GloSea6) model [33] to generate daily 3+1 ensemble/control forecasts for 60-day lead time.

**NCEP**. The National Centers for Environmental Prediction uses the Climate Forecast System 2 (CFSv2) model [34] to generate daily 15+1 ensemble/control forecast for 45-day lead time.

**CMA**. The China Meteorological Administration uses the Beijing Climate Center (BCC) fully-coupled BCC-CSM2-HR model [35] to generate 3+1 ensemble/control forecasts at 3-day interval for 60-day lead time.

**ECMWF**. The European Centre for Medium-Range Weather Forecasts uses the operational Integrated Forecasting System (IFS) that includes advanced data assimilation strategies and global numerical model of the Earth system [36]. In particular, we use the CY41R1 version of the IFS to generate 50+1 ensemble/control forecasts twice weekly for 46-day lead time.

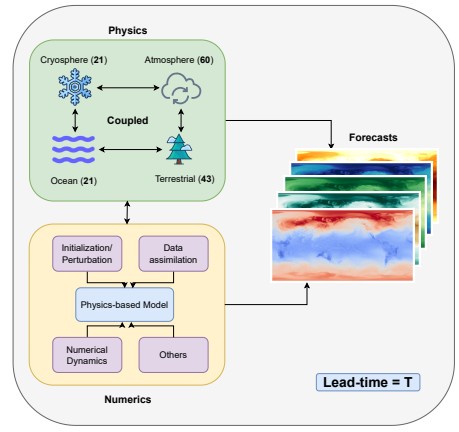

Figure 2: Physics-based simulations that couple different parts of the Earth system along with their operational choices such as data assimilation. The brackets are the number of variables provided in ChaosBench.

## 3.3 Auxiliary

In addition to baseline forecasts from physics-based and data-driven models, we provide additional auxiliary data and baselines. This includes **climatology**, the long-term weather-state statistics, and **persistence**, which uses initial observation for subsequent rollouts.

# 4 Benchmark Metrics

We provide an assortment of metrics, which we divide into deterministic, probabilistic, and several proposed physics-based criteria, for increased explainability. For each metric, unless otherwise noted, we apply a weighting scheme at each latitude $\theta_i$ as defined by Equation 1.

$$w(\theta_i) = \frac{cos(\theta_i)}{\frac{1}{|\boldsymbol{\theta}|} \sum_{a=1}^{|\boldsymbol{\theta}|} cos(\theta_a)} \tag{1}$$

where $\boldsymbol{\theta}$ is the set of all latitudes in our data, and $|\boldsymbol{\theta}|$ is its cardinality. We denote the input at time $t$ as $\mathbf{X}_t \in \mathbb{R}^{h \times w \times p}$, where $h, w, p$ represent the height (i.e., latitude), width (i.e., longitude), and parameter (e.g., temperature) with its associated vertical level (e.g., 1000-hpa or surface). In addition, we denote $\{\mathbf{Y}_t, \hat{\mathbf{Y}}_t\} \in \mathbb{R}^{h \times w \times p}$ as the ground-truth label and prediction respectively. Finally, we denote each element of latitude and longitude as $\theta_i \in \boldsymbol{\theta}$ and $\gamma_j \in \boldsymbol{\gamma}$.

## 4.1 Deterministic Metrics

We provide popular deterministic metrics in the machine learning and climate science literature alike, including RMSE, Bias, ACC, and MS-SSIM.

**Root Mean Squared Error (RMSE)** is useful to penalize outliers, which are especially critical for weather and climate applications such as extreme event prediction (Equation S1).

**Bias** assists us to identify misspecification and systematic errors present in the model (Equation S2).

**Anomaly Correlation Coefficient (ACC)** measures the correlation between predicted and observed anomalies. This metric is especially useful in weather and climate applications, where deviations from the norm (e.g., temperature anomalies) often reveal interesting insights (Equation S3).

**Multi-Scale Structural Similarity (MS-SSIM)** [37] compares structural similarity between forecast and ground-truth label across scales (refer to Appendix F.1.4 for more details). This is especially useful in weather systems because they occur at multiple scales, from large systems like cyclones, to smaller features like localized rain thunderstorms.

## 4.2 Physics Metrics

As illustrated in Figure 3, we find that in general, data-driven forecasts tend to become blurry (Figure 3a) due to power divergence in the spectral domain (Figure 3b + S10). This motivates us to propose two physics-based metrics that measure the deviation or difference between the power spectra of prediction $\hat{S}(k)$ and target $S(k)$, where $k \in \mathbf{K}$, and $\mathbf{K}$ is the set of all scalar wavenumbers from 2D Fourier transform. Focusing on high-frequency components, we introduce $\mathbf{K}_q = \{k \in \mathbf{K} \mid k \geq Q(q)\}$, where $Q$ is the quantile function of $\mathbf{K}$ and $q \in [0, 1]$. We set $q = 0$ or $q = 0.9$ for training and evaluation respectively. We denote $S_q = \{S(k) \mid k \in \mathbf{K}_q\}$ as the corresponding power spectra on $\mathbf{K}_q$, and we normalize the distribution to $S'(k)$ such that it sums up to 1. Similarly we use $\hat{S}'(k)$ to denote the normalized power for predictions.

**Spectral Divergence (SpecDiv)** follows principles from Kullback–Leibler (KL) divergence [38] where we compute the expectation of the log ratio between target $S'(k)$ and prediction $\hat{S}'(k)$ spectra, and is defined in Equation 2 (see Listing S1 for PYTORCH psuedocode).

$$\mathcal{M}_{SpecDiv} = \sum_k S'(k) \cdot \log(S'(k)/\hat{S}'(k)) \tag{2}$$

**Spectral Residual (SpecRes)** follows principles from RMSE and adapted from [39] where we compute the root of the expected squared residual, and is defined in Equation 3 (see Listing S2 for PYTORCH psuedocode).

$$\mathcal{M}_{SpecRes} = \sqrt{\mathbb{E}_k[(\hat{S}'(k) - S'(k))^2]} \tag{3}$$

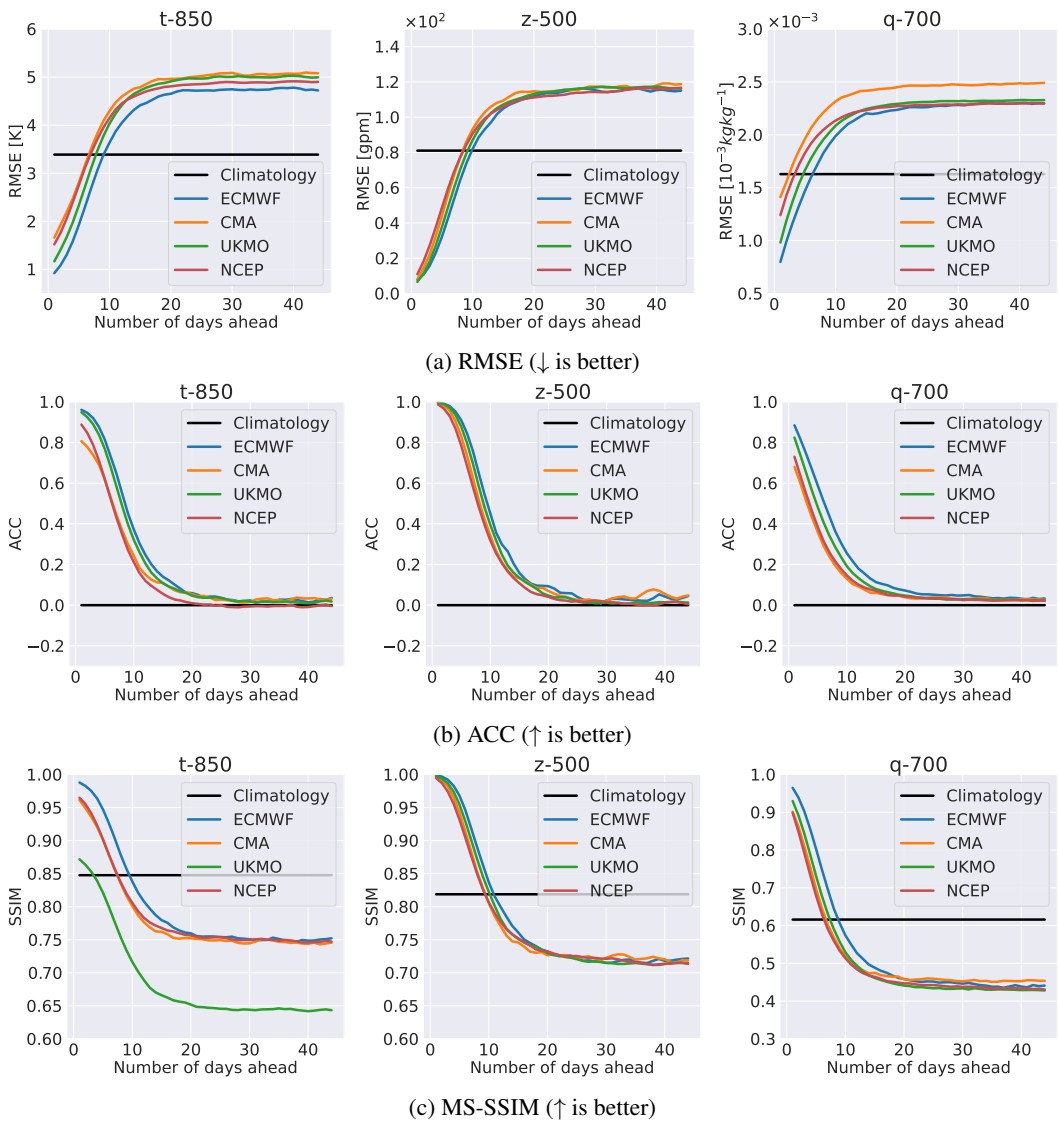

(a) RMSE (↓ is better)

(b) ACC (↑ is better)

(c) MS-SSIM (↑ is better)

Figure 4: Evaluation results between baseline climatology (black line) and physics-based control/deterministic forecasts. At longer forecasting horizon, most physics-based control/deterministic forecasts perform worse than climatology.

The expectations are calculated over $\mathbf{K}_q$. For both physics-based metrics, the value will be zero if the power spectra of the forecast is identical to the target, but will increase as discrepancy emerges. Essentially, both metrics measure how well the forecasts *preserve* signals across the frequency spectrum.

### 4.3 Probabilistic Metrics

In addition to the probabilistic version of RMSE, Bias, ACC, MS-SSIM, SpecDiv, and SpecRes where we take their expectation with respect to the ensemble members (Equations S14-S19), we also use several probabilistic metrics to evaluate ensemble forecasts critical for long-range S2S prediction.

**Continuous Ranked Probability Score (CRPS)** evaluates the accuracy of the ensemble distribution against the target. Low CRPS values require forecasts to be reliable, where the predicted uncertainty aligns with the actual uncertainty, and a smaller uncertainty is preferable (Equation S20).

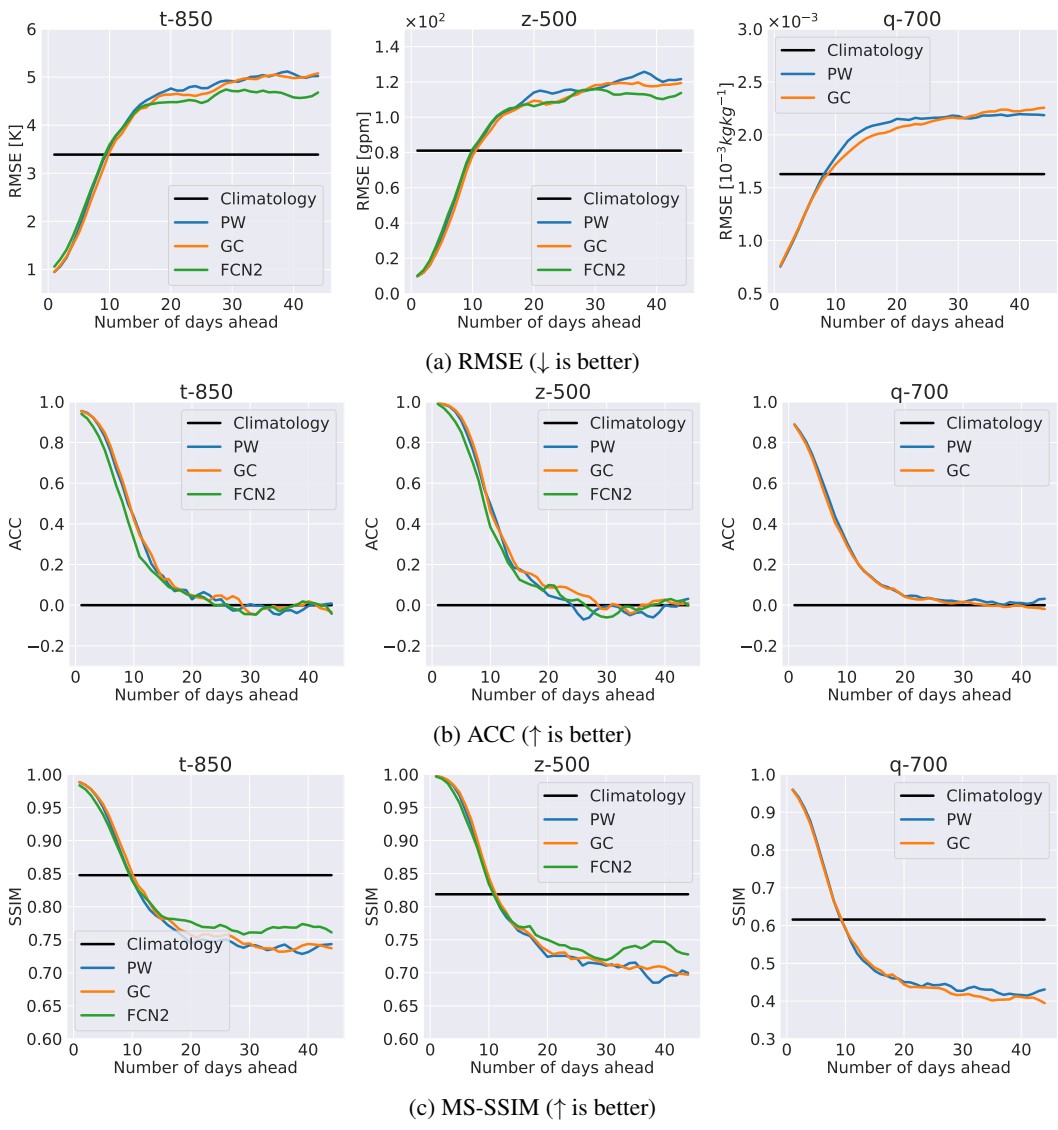

(a) RMSE (↓ is better)

(b) ACC (↑ is better)

(c) MS-SSIM (↑ is better)

Figure 5: Evaluation results between baseline climatology (black line) and data-driven models including PanguWeather (PW), GraphCast (GC), and FourCastNetV2 (FCN2). We find that deterministic ML models perform worse than climatology on S2S timescale. Note: FCN2 lacks q-700.

**Continuous Ranked Probability Skill Score (CRPSS)** evaluates the skill of probabilistic forecast relative to climatology variability; CRPSS > 0 suggests skillfulness and vice versa (Equation S21).

**Spread** quantifies the uncertainty in ensemble forecasts by measuring the variability among ensemble members, which helps to understand the range of possible outcomes and confidence (Equation S22).

**Spread/Skill Ratio** balances the ensemble spread with the forecast skill (e.g., RMSE); ideally, a well-calibrated ensemble should have a spread that matches the forecast skill (Equation S23).

## 5 Benchmark Results

Throughout this section, we report headline results on $\hat{\mathbf{X}} \in \{\text{t-850, z-500, q-700}\}$, following Weatherbench v2 [40]. The full benchmark scores are available at https://leap-stc.github.io/ChaosBench. We primarily use four state-of-the-art models for comparison including ViT/ClimaX [17], PanguWeather [18], GraphCast, and FourCastNetV2 [9] [7]. However, whenever ablation is performed, we use popular baselines including Lagged Autoencoder [41], ResNet [42], UNet [22], and FNO [43] trained

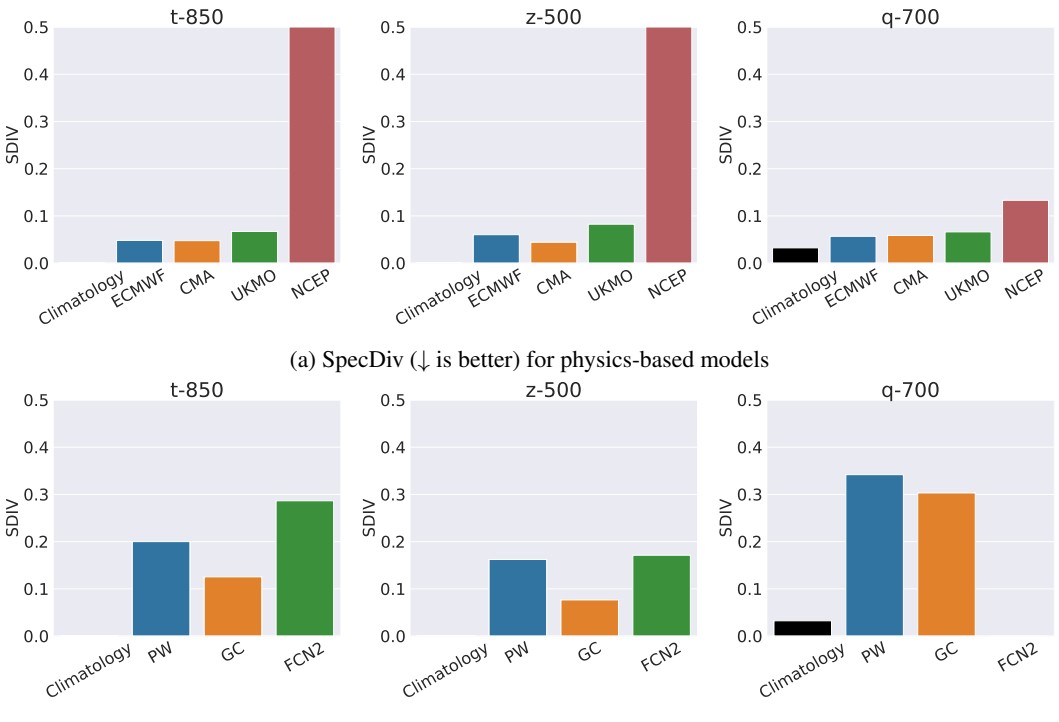

(a) SpecDiv (↓ is better) for physics-based models

(b) SpecDiv (↓ is better) for data-driven models

Figure 6: Spectral divergence between (a) physics-based, and (b) data-driven models. Overall, we observe that the latter perform worse than their physics-based counterpart (barring NCEP) on time-averaged spectral divergence. Note: FCN2 lacks q-700.

on 1979-2015 data and validated on 2016-2021 data. All evaluations presented here are done on the held-out 2022 data. The full implementation details are discussed in Appendix E.

Table 2: Performance metrics for SoTAs with different training strategies, at $\Delta t = 44$

| Metrics | Variables | Reference | Autoregressive | | | Direct |
|---|---|---|---|---|---|---|
| | | **Climatology** | **PW** | **GC** | **FCN2** | **ViT/ClimaX** |
| **RMSE ↓** | t-850 (K) | 3.39 | 5.85 | 5.87 | 5.11 | **3.56** |
| | z-500 (gpm) | 81.0 | 120.9 | 136.0 | 112.4 | **83.1** |
| | q-700 ($\times 10^{-3}$) | 1.62 | 2.35 | 2.28 | - | **1.66** |
| **MS-SSIM ↑** | t-850 | 0.85 | 0.70 | 0.70 | 0.74 | **0.83** |
| | z-500 | 0.82 | 0.68 | 0.66 | 0.72 | **0.81** |
| | q-700 | 0.62 | 0.43 | 0.45 | - | **0.59** |
| **SpecDiv ↓** | t-850 | 0.01 | 0.25 | **0.05** | 0.28 | 0.20 |
| | z-500 | 0.01 | 0.33 | **0.03** | 0.11 | 0.13 |
| | q-700 | 0.03 | **0.23** | 0.27 | - | 0.28 |

**Collapse in Predictive Skill**. As shown in Figure 4 (+ S2), control forecasts from various operational centers perform worse than climatology at the S2S scale beyond 15 days. A similar phenomenon of skill collapse is evident in data-driven models, as depicted in Figure 5 (+ S3). Unlike their physics-based counterparts, these forecasts exhibit significantly higher spectral divergence as evidenced in Figure 6, indicating low predictive skill for multi-scale structures over long rollouts. This leads to the blurring artifacts previously discussed. The pervasive lack of predictive skill underscores the notoriously difficult challenge of S2S forecasting and highlights huge potential for improvement.

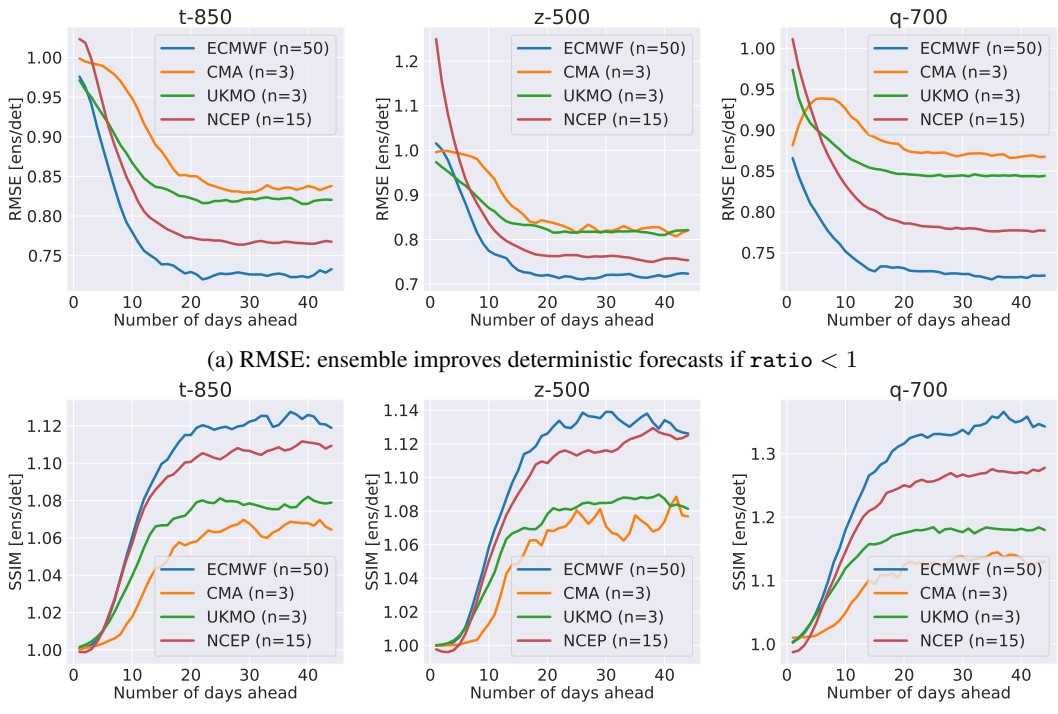

(a) RMSE: ensemble improves deterministic forecasts if `ratio` $< 1$

(b) MS-SSIM: ensemble improves deterministic forecasts if `ratio` $> 1$

Figure 7: Metrics ratio e.g., $\text{RMSE}_{ens}/\text{RMSE}_{det}$ between ensemble and deterministic forecasts, where the former improves the latter by accounting for IC uncertainty that can lead to trajectory divergences. Note: $n$ represents the number of ensemble members.

**Ensemble Forecasts Account for IC Uncertainty**. Despite the underperformance of deterministic models, many studies have highlighted the potential of ensemble forecasts to account for trajectory divergences caused by IC uncertainties [44, 45, 46], also known as the butterfly effect [15]. Figure S4 shows that the performance of ensembles across physics-based models improves relative to their deterministic counterparts. For instance, when we take the metrics ratio between ensemble and deterministic forecasts as in Figure 7 (+ S5), the ratio of RMSE decreases with lead time, while the ratio of MS-SSIM improves over time with little significant changes in SpecDiv. The extent of improvement also appears to be affected by the number of ensemble members i.e., higher ensemble size $n$ appears to improve skillfulness. We also note similar insights from data-driven ensembling strategy as discussed in Section G.3. This highlights the importance of building a well-dispersed ensemble that accounts for long-range divergences for improved S2S predictability.

**Minimizing Error Propagation Promotes Stability**. Different training and inference strategies have been proposed to improve the accuracy and stability of data-driven weather emulators. Chief among these are the autoregressive and direct approaches [47]. The former iteratively cycles through small interval to reach the target lead-time i.e., $\Delta t = N\delta t$ where $N \in \mathbb{Z}^+$ is the number of such compositions, while the latter directly outputs $\Delta t$. As summarized in Table 2, we find models trained directly (e.g., ViT/ClimaX) have better performance than those used autoregressively (e.g., PW, GC, FCN2). This suggests that error propagation is a significant source of error, and controlling for stability is key to extend the predictability range of weather emulators. Once stability is achieved, the remaining sources of errors including uncertainties in observation and/or modeling framework can be improved through more data, better model, or both through data assimilation for instance [48].

**Physical Constraints Yield Improved Performance**. We find models that explicitly incorporate physical knowledge (e.g., learning spectral signals beyond pixel information) have better performance across metrics, such as FNO, as summarized in Table S8 given identical parameter budget of $10^6$. This phenomena is unsurprising and has been repeatedly demonstrated in many real-world applications of physics-informed deep learning, for instance.

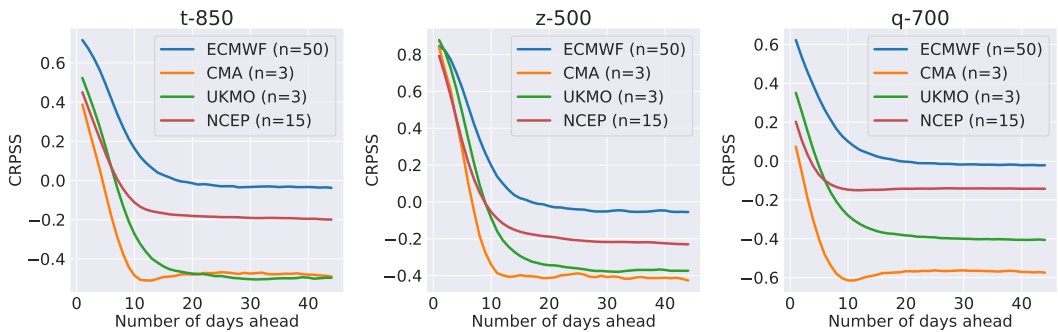

Figure 8: Probabilistic evaluation on ensemble forecasts indicating current skill limits of 15-20 days; CRPSS $> 0$ suggests skills better than climatology variability. Note: $n$ represents the number of ensemble members.

**Current Limits of S2S Predictability**. Given our best models, we evaluate the extent of predictability in order to base our next steps. As illustrated in Figure 8 (+ S6), we find that ECMWF high-resolution ensemble, dubbed as the gold standard, still has the best performance in terms of CRPSS (vs ERA5 climatology), with a predictability range of around 15-20 days ahead before its skill collapses to climatology (i.e., CRPSS $\rightarrow 0$). However, the resurgence of data-driven models are rapidly transforming the field as they are able to efficiently distil knowledge and automatically discover emergent patterns from large-scale, high-dimensional dataset, instead of first reducing them to physical functions with limited set of variables requiring constant calibration as is traditionally done in NWPs. The challenge, therefore, is to extend the predictability range of weather system as a representation of large-scale chaos, and we welcome the machine learning communities to take part in this open effort.

## 6 Conclusion

We present ChaosBench, a challenging benchmark to extend the predictability range of weather emulators into the S2S timescale where many processes with significant socioeconomic repercussions tend to occur, including extreme events. In addition to providing diverse datasets beyond ERA5 for a full Earth system emulation, we also perform extensive benchmarking on state-of-the-art data-driven and physics-based models alike. Through various ablation, we systematically find that skillfulness can be extended by ensemble forecasting, controlling for exponential error growth, and incorporating physical knowledge in our modeling approaches.

**Future Work**. Our input datasets have relatively coarse spatiotemporal resolution to match that of physics-based S2S forecasts. Nevertheless, we make the data processing pipeline open-source, allowing users to easily process inputs of the desired resolution (see Section B.4 for more details). We are planning for a multi-source reanalysis products (e.g., MERRA-2 [49]), leveraging diverse dataset strengths, such as the assimilation of different set of observations. As always, we welcome any contribution from the open-source community to solve this important yet understudied problem. And any comments, feedback, and/or future feature requests can be directed to the corresponding author or through the Github issue tracker at https://github.com/leap-stc/ChaosBench.

## Acknowledgments and Disclosure of Funding

We would like to thank Matthew Wilson, Tom Andersson, and Dale Durran for the insightful discussion during the earlier version of the manuscript. The authors also acknowledge funding, computing, and storage resources from the NSF Science and Technology Center (STC) Learning the Earth with Artificial Intelligence and Physics (LEAP) (Award #2019625) and the Department of Energy (DOE) Advanced Scientific Computing Research (ASCR) program (DE-SC0022255). AG would like to acknowledge support from Google and Schmidt Sciences. Last but definitely not least, we acknowledge the comprehensive S2S database emerging from the joint initiative of the World Weather Research Programme (WWRP) and the World Climate Research Programme (WCRP). The original S2S database is hosted at ECMWF as an extension of the TIGGE database.

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
