# ChaosBench: A Multi-Channel, Physics-Based Benchmark for Subseasonal-to-Seasonal Climate Prediction

## Supplementary Material

**Juan Nathaniel**[1,*], **Yongquan Qu**[1], **Tung Nguyen**[2], **Sungduk Yu**[3,5], **Julius Busecke**[1,4],
**Aditya Grover**[2], **Pierre Gentine**[1]
[1]Columbia University, [2]UCLA, [3]UCI, [4]LDEO, [5] Intel Labs

## A   Accountability and Reproducibility Statement

ChaosBench is published under the open source GNU General Public License. Further development and potential updates discussed in the limitations section will take place on the ChaosBench page. Furthermore, we are committed to maintaining and preserving the ChaosBench benchmark. Ongoing maintenance also includes tracking and resolving issues identified by the broader community after release. User feedback will be closely monitored via the GitHub issue tracker. All assets are hosted on GitHub and HuggingFace, which guarantees reliable and stable storage.

**Dataset**: All our dataset, present and future (e.g., with more years, multi-resolution support, etc) are available at https://huggingface.co/datasets/LEAP/ChaosBench.

**Model Checkpoints**: All of our model checkpoints used for the purposes of ablation in this work are available at https://huggingface.co/datasets/LEAP/ChaosBench/tree/main/logs.

**Code**: Our code and its future extension based on community feedback is accessible at https://github.com/leap-stc/ChaosBench.

**Documentation**: Finally, our main webpage will keep track of all important updates and latest documentation, and is accessible at https://leap-stc.github.io/ChaosBench.

---

*Corresponding author: jn2808@columbia.edu

# B Getting Started

Here, we provide a detailed description on how to prepare the necessary data, perform training, and benchmark your own model. However, we refer users to our webpage `https://leap-stc.github.io/ChaosBench` for the most updated how-to guides.

The following sections assume successful cloning of our `Github` repository `https://github.com/leap-stc/ChaosBench`. If you find any problems, feel free to contact us or raise an issue.

## B.1 Data Preparation

**First**, navigate to the repository directory and install the necessary dependencies.

```
$ cd ChaosBench
$ pip install -r requirements.txt
```

**Second**, download the dataset using the following commands.

```
$ cd data/
$ wget https://huggingface.co/datasets/LEAP/ChaosBench/resolve/main/
    process.sh
$ chmod +x process.sh
```

**Third**, process the following required and optional dataset.

```
# Required for inputs and climatology (e.g., normalization)
$ ./process.sh era5
$ ./process.sh lra5
$ ./process.sh oras5
$ ./process.sh climatology

# Optional: control (deterministic) forecasts
$ ./process.sh ukmo
$ ./process.sh ncep
$ ./process.sh cma
$ ./process.sh ecmwf

# Optional: perturbed (ensemble) forecasts
$ ./process.sh ukmo_ensemble
$ ./process.sh ncep_ensemble
$ ./process.sh cma_ensemble
$ ./process.sh ecmwf_ensemble

# Optional: SoTa (deterministic) forecasts
$ ./process.sh panguweather
$ ./process.sh graphcast
$ ./process.sh fourcastnetv2
```

## B.2   Training

We will cover how training can generally be performed, followed by how one can switch between different training strategies by manipulating the config `.yaml` file.

**First**, define your model class.

```
# An example can be found for e.g. <YOUR_MODEL> == fno

$ touch chaosbench/models/<YOUR_MODEL>.py
```

**Second**, import and initialize your model in the main `chaosbench/models/model.py` file, given the pseudocode below.

```
# Examples for lagged_ae, fno, resnet, unet are provided

import lightning.pytorch as pl
from chaosbench.models import YOUR_MODEL

class S2SBenchmarkModel(pl.LightningModule):

    def __init__(
        self,
        ...
    ):
        super(S2SBenchmarkModel, self).__init__()

        # Initialize your model
        self.model = YOUR_MODEL.BEST_MODEL(...)

        # The rest of model construction logic
```

**Third**, run the `train.py` script. We recommend using GPUs for training.

```
# The _s2s suffix identifies data-driven models

$ python train.py --config_filepath chaosbench/configs/<YOUR_MODEL>
    _s2s.yaml
```

Now you will notice that there is a `.yaml` file. We define the definition of each field, allowing for greater control over different training strategies.

```
# The .yaml file always has two sections: model_args and data_args

model_args:
    model_name: <str>        # Name of your model e.g., 'unet_s2s'
    input_size: <int>        # Input size, default: 60 (ERA5)
    output_size: <int>       # Output size, default: 60 (ERA5)
    learning_rate: <float>   # Learning rate
    num_workers: <int>       # Number of workers
    epochs: <int>            # Number of epochs
    t_max: <int>             # Learning rate scheduler
    only_headline: <bool>    # Only optimized for config.HEADLINE_VARS

data_args:
    batch_size: <int>        # Batch size
    train_years: [...]       # Train years e.g., [1979, ...]
    val_years: [...]         # Val years e.g., [2016, ...]
    n_step: <int, 1>         # Number of autoregressive training steps
    lead_time: <int, 1>      # N-day ahead forecast (for direct scheme)
    land_vars: [...]         # Extra LRA5 vars e.g., ['t2m', ...]
    ocean_vars: [...]        # Extra ORAS5 vars e.g., ['sosstsst', ...]
```

Note,

1. If `only_headline` is set to `True`, then the model is optimized only for a subset of variables defined in `config.HEADLINE_VARS` (default: `False`).

2. If `n_step` is set to values greater than 1, the models will train over $n$-autoregressive steps (default: 1).

3. If `lead_time` is set to values greater than 1, the models will be able to forecast $n$-days ahead. For example, in our direct forecasts, if `lead_time` is set to 4, our model will predict the states 4 days into the future (default: 1).

4. If `land_vars` and/or `ocean_vars` are set with entries from the acronyms in Tables S3 and S2, these will be used as additional inputs and targets, on top of `ERA5` variables (default: []).

## B.3 Evaluation

Once training is done, we can perform evaluation depending on the use case. We recommend using GPUs for evaluation.

**First**, if we have an autoregressive model, we can simply run:

```
# Evaluating autoregressive model, e.g.,
# --model_name 'unet_s2s'
# --eval_years 2022
# --version_num 0        ## Checkpoint versions autogenerated in logs/
# --lra5 't2m' 'tp'      ## Additional LRA5 vars to be evaluated
# --oras5 'sosstsst'     ## Additional ORAS5 vars to be evaluated

$ python eval_iter.py --model_name <str> --eval_years <int> --
    version_num <int> --lra5 [...] --oras5 [...]
```

**Second**, if we have a collection of models trained specifically for unique `lead_time`, we can run:

```
# Evaluating direct model with the default sequence of
# lead_time = [1, 5, 10, 15, 20, 25, 30, 35, 40, 44] e.g.,
# --model_name 'unet_s2s'
# --eval_years 2022
# --version_nums 0 4 5 6 7 8 9 10 11 12
# --lra5 't2m' 'tp'      ## Additional LRA5 vars to be evaluated
# --oras5 'sosstsst'     ## Additional ORAS5 vars to be evaluated

$ python eval_direct.py --model_name <str> --eval_years <int> --
    version_nums [...] --lra5 [...] --oras5 [...]
```

**Third**, if we have a probabilistic model that generates ensemble forecasts (e.g., one checkpoint represents one ensemble member) and are supposed to be evaluated with additional probabilistic metrics, we can run:

```
# Evaluating ensembles with additional probabilistic metrics e.g.,
# --model_name 'unet_ensemble_s2s'
# --eval_years 2022
# --version_nums 0 1 2  ## One ensemble member per version
# --lra5 't2m' 'tp'      ## Additional LRA5 vars to be evaluated
# --oras5 'sosstsst'     ## Additional ORAS5 vars to be evaluated

$ python eval_ensemble.py --model_name <str> --eval_years <int> --
    version_nums [...] --lra5 [...] --oras5 [...]
```

## B.4 Optional: Processing Multi-Resolution Input

We open-source the data processing script to allow users to process the inputs given different resolution (highest is 0.25-degree):

```
# Process inputs with e.g., 0.25-degree resolution
$ python scripts/process_atmos.py --resolution 0.25 # ERA5
$ python scripts/process_ocean.py --resolution 0.25 # ORAS5
$ python scripts/process_land.py  --resolution 0.25 # LRA5
```

## C    Related Work

Here we discuss the criteria used to compare different S2S benchmark. This list is by no means exhaustive and there exists many ways to interpret the different contribution, strength, and scope of each. We refer interested reader to the respective benchmark paper and website.

**On Input Variables**. The number of input channels indicates the number of *unique* variables used for training data-driven models. For instance, in the case of SubseasonalClimateUSA, these include tmin, tmax, tmean, precip_agg, precip_mean, SST, SIC, z-10, z100, z500, z850, u-250, u-925, v-250, v-925, surface_P, RH, SSP, precipitable water, PE, DEM, KG, MJO-phase, MJO-amp, ENSO-I, despite them having similar (25) variables across data sources.

**On Target Variables and Agencies**. Similarly, the number of target channels represent the variables these benchmarks are aiming for. This is closely related to the number of benchmark agencies, which refers to the number of physics-based simulations used as target, rather than inputs. In the case for SubseasonalClimateUSA, for instance, the number of target channels correspond to two: precipitation and surface temperature, while the number of benchmark agencies is also two: CFSv2 (NCEP) and IFS (ECMWF), despite them using multiple other simulations generated from agencies but as inputs; though evaluated on all in their follow-up work [50] despite not initially described in the dataset paper.

**On Physics Metrics**. The flag for physics-based metrics indicates whether these benchmarks incorporate not just physical explanation, but also formulate them as scalar and differentiable metrics for future optimization problem.

**On Probabilistic Metrics**. The flag for probabilistic metrics indicates whether these benchmarks incorporate probabilistic (e.g., CRPS, CRPSS, Spread, SSR), in addition to deterministic metrics.

**On Spatial Extent**. Furthermore, the spatial extent indicates the extent of the target benchmark, rather than of the input dataset. This is because some of the more challenging S2S forecasting task is to get the correct global space-time correlation, and having a full global coverage provides a more complete evaluation.

# D ChaosBench

## D.1 Observations from Reanalysis Products

### D.1.1 ERA5

The following table indicates the 48 variables that are inferred by physics-based models. Note that the Input ERA5 observations contains **ALL** fields, including the unchecked boxes:

Table S1: List of ERA5 reanalysis variables

| Parameters/Levels (hPa) | 1000 | 925 | 850 | 700 | 500 | 300 | 200 | 100 | 50 | 10 |
|---|---|---|---|---|---|---|---|---|---|---|
| Geopotential height, z ($gpm$) | ✓ | ✓ | ✓ | ✓ | ✓ | ✓ | ✓ | ✓ | ✓ | ✓ |
| Specific humidity, q ($kg\,kg^{-1}$) | ✓ | ✓ | ✓ | ✓ | ✓ | ✓ | ✓ | | | |
| Temperature, t ($K$) | ✓ | ✓ | ✓ | ✓ | ✓ | ✓ | ✓ | ✓ | ✓ | ✓ |
| U component of wind, u ($ms^{-1}$) | ✓ | ✓ | ✓ | ✓ | ✓ | ✓ | ✓ | ✓ | ✓ | ✓ |
| V component of wind, v ($ms^{-1}$) | ✓ | ✓ | ✓ | ✓ | ✓ | ✓ | ✓ | ✓ | ✓ | ✓ |
| Vertical velocity, w ($Pas^{-1}$) | | | | | ✓ | | | | | |

### D.1.2 ORAS5

The variables for ORAS5 consist of the following as described in Table S2.

Table S2: List of ORAS5 reanalysis variables

| Acronyms | Long Name | Units |
|---|---|---|
| iicethic | sea ice thickness | m |
| iicevelu | sea ice zonal velocity | $ms^{-1}$ |
| iicevelv | sea ice meridional velocity | $ms^{-1}$ |
| ileadfra | sea ice concentration | (0-1) |
| so14chgt | depth of 14° isotherm | m |
| so17chgt | depth of 17° isotherm | m |
| so20chgt | depth of 20° isotherm | m |
| so26chgt | depth of 26° isotherm | m |
| so28chgt | depth of 28° isotherm | m |
| sohefldo | net downward heat flux | $Wm^{-2}$ |
| sohtc300 | heat content at upper 300m | $Jm^{-2}$ |
| sohtc700 | heat content at upper 700m | $Jm^{-2}$ |
| sohtcbtm | heat content for total water column | $Jm^{-2}$ |
| sometauy | meridionial wind stress | $Nm^{-2}$ |
| somxl010 | mixed layer depth 0.01 | m |
| somxl030 | mixed layer depth 0.03 | m |
| sosaline | salinity | PSU |
| sossheig | sea surface height | m |
| sosstsst | sea surface temperature | $°C$ |
| sowaflup | net upward water flux | $kg/m^2/s$ |
| sozotaux | zonal wind stress | $Nm^{-2}$ |

## D.1.3 LRA5

The variables for LRA5 consist of the following as described in Table S3.

Table S3: List of LRA5 reanalysis variables

| Acronyms | Long Name | Units |
|---|---|---|
| asn | snow albedo | (0 - 1) |
| d2m | 2-meter dewpoint temperature | K |
| e | total evaporation | m of water equivalent |
| es | snow evaporation | m of water equivalent |
| evabs | evaporation from bare soil | m of water equivalent |
| evaow | evaporation from open water | m of water equivalent |
| evatc | evaporation from top of canopy | m of water equivalent |
| evavt | evaporation from vegetation transpiration | m of water equivalent |
| fal | forecaste albedo | (0 - 1) |
| lai_hv | leaf area index, high vegetation | $m^2m^{-2}$ |
| lai_lv | leaf area index, low vegetation | $m^2m^{-2}$ |
| pev | potential evaporation | m |
| ro | runoff | m |
| rsn | snow density | $kgm^{-3}$ |
| sd | snow depth | m of water equivalent |
| sde | snow depth water equivalent | m |
| sf | snowfall | m of water equivalent |
| skt | skin temperature | K |
| slhf | surface latent heat flux | $Jm^{-2}$ |
| smlt | snowmelt | m of water equivalent |
| snowc | snowcover | % |
| sp | surface pressure | Pa |
| src | skin reservoir content | m of water equivalent |
| sro | surface runoff | m |
| sshf | surface sensible heat flux | $Jm^{-2}$ |
| ssr | net solar radiation | $Jm^{-2}$ |
| ssrd | download solar radiation | $Jm^{-2}$ |
| ssro | sub-surface runoff | m |
| stl1 | soil temperature level 1 | K |
| stl2 | soil temperature level 2 | K |
| stl3 | soil temperature level 3 | K |
| stl4 | soil temperature level 4 | K |
| str | net thermal radiation | $Jm^{-2}$ |
| strd | downward thermal radiation | $Jm^{-2}$ |
| swvl1 | volumetric soil water layer 1 | $m^3m^{-3}$ |
| swvl2 | volumetric soil water layer 2 | $m^3m^{-3}$ |
| swvl3 | volumetric soil water layer 3 | $m^3m^{-3}$ |
| swvl4 | volumetric soil water layer 4 | $m^3m^{-3}$ |
| t2m | 2-meter temperature | K |
| tp | total precipitation | m |
| tsn | temperature of snow layer | K |
| u10 | 10-meter u-wind | $ms^{-1}$ |
| v10 | 10-meter v-wind | $ms^{-1}$ |

### D.2 Physics-Based Simulations

In this section, we describe in detail the physics-based models used as baselines in ChaosBench. Wherever possible, we discuss specific strategies regarding coupling to the ocean, sea ice, wave, land, initialization and perturbation strategies, specifications of initial/boundary conditions, as well as other numerical considerations to generate forecast.

#### D.2.1 The UK Meteorological Office (UKMO) [33]

- **Initialization and Ensemble**. The UKMO model employs the lagged initialization strategy to generate an ensemble of forecasts (4 in this case) at different initialization time to improve prediction stability.

- **Coupling with ocean** is performed with the Global Ocean 6.0 model [51], based on NEMO3.6 [52] with 0.25 degree horizontal resolution and 75 vertical pressure levels. The ocean model is initialized and calibrated using Nonlinear Evolutionary Model VARiation (NEMOVAR) [53], a specific data assimilation strategy that uses temperature, salinity profiles, altimeter-derived sea level anomalies to calibrate forecasts. Frequency of coupling is 1-hourly.

- **Coupling with sea ice** is performed with the Global Sea Ice 8.1 (CICE5.1.2) model [54], and again initialized from NEMOVAR.

- **Coupling with wave model** is not yet operational.

- **Coupling with land surface** is performed with the Joint UK Land Environment Simulator (JULES) [55]. Soil moisture, soil temperature, and snow are initialized using JULES and forced using the the Japanese 55-year Reanalysis (JRA-55) data [56]. The land surface model is paramaterized by land cover type from a combination of satellite (e.g., MODIS LAI [57]) and radiometer data (e.g., AVHRR [58]). In addition, another parameterization in the form of soil characteristics is derived from the Harmonized World Soil Database [59].

- **Model grid** uses the Arakawa C-grid [60] to solve partial differential equations on a spherical surface. In particular, the velocity components (such as zonal and meridional wind) are defined at the center of each face of the grid cells (in the case of a rectilinear grid) or along cell edges (in the case of a curvilinear grid). The scalar quantities such as pressure or temperature are computed at the corners of the grid cells.

- **Large-scale dynamics** uses the Semi-Lagrangian approach. It does not strictly follow fluid parcels (i.e., Lagrangian), but it does calculate the value of a field, such as temperature (i.e., Eulerian) by tracing back along the trajectory that a fluid parcel would have taken to reach a specific point at the current time step. This backward trajectory is used to find the origin of the fluid parcel and determine its properties, which are then used to update the model fields. This hybrid approach is therefore termed Semi-Lagrangian.

#### D.2.2 National Centers for Environmental Prediction (NCEP) [61]

- **Initialization and Ensemble**. The NCEP model adds small perturbation to the atmospheric, oceanic and land analysis at each cycle across 4 ensemble to reduce sensitivity to initial conditions.

- **Coupling with ocean** is performed with the GFDL Modular Ocean Model version 4 (MOM4) model that has a spatial resolution of 0.5-degree and 0.25-degree in the longitude-latitude directions [62]. There are 40 vertical pressure levels.

- **Coupling with sea ice** is also performed with the GFDL Sea Ice Simulator (SIS), which models the thermodynamics and overall dynamics of sea ice [62].

- **Coupling with wave model** is not yet operational.

- **Coupling with land surface** is performed with 4-layer Noah Land surface model 2.7.1 [63]. Soil moisture, soil temperature, and snow are initialized using Noah and forced using the Climate Forecast System [61] and the Global Land Data Assimilation System [64] reanalysis data. The land surface model is parameterized by land cover type AVHRR data. In addition, another paramaterization in the form of soil characteristics is derived from the world soil climate database [65].

- **Model grid** uses the Gaussian grid [66], where the longitude (x-axis) are evenly spaced while the latitudes (y-axis) are not. Instead, they are determined by the roots of the associated Legendre polynomials, which correspond to the Gaussian quadrature points for the sphere. This ensures that the actual area represented by each grid cell is more uniform.

- **Large-scale dynamics** uses the Spectral approach. It solves partial differential equations by transforming them from the physical space into the spectral domain. In the latter case, the equations are transformed into a series of coefficients that represent the amplitude of waves across scales. The transformations are usually done using Fourier series for periodic domains or spherical harmonics when dealing with the whole Earth's surface [66]. This method is especially beneficial for smooth functions and for representing large-scale wave phenomena, such as the Rossby waves, which are important for understanding weather and climate.

### D.2.3 China Meteorological Administration (CMA) [35]

- **Initialization and Ensemble**. The CMA model uses the lagged average forecasting (LAF) method across 4 ensemble members to ensure that the mean forecast is less sensitive to initial conditions.

- **Coupling with ocean** is performed with the GFDL MOM4 model, which has 40 vertical pressure levels [62]. Frequency of coupling is 2-hourly.

- **Coupling with sea ice** is performed with the GFDL Sea Ice Simulator (SIS), similar to that used by NCEP [62].

- **Coupling with wave model** is not yet operational.

- **Coupling with land surface** is performed with the Atmosphere-Vegetation Interaction Model version 2 (AVIM2) model [67] and the NCAR NCAR Community Land Model version 3.0 (CLMv3) [68]. Soil moisture, soil temperature, and snow are not initialized directly using reanalysis data, as used by other land surface models. Rather, air-sea-land-ice coupled model is forced by near-surface atmospheric and ocean reanalysis in a long-term integration, and the land initial conditions are produced as a by-product. As a result, the parameterization of land cover type is done by this process, while soil characteristics is derived from the Harmonized World Soil Database [59].

- **Model grid** uses the Gaussian grid [66], similar to that used by the NCEP.

- **Large-scale dynamics** uses a mixture of Spectral approach for the vorticity, temperature, and surface pressure, as well as Semi-Lagrangian for specific humidity and cloud waters other tracers.

### D.2.4 European Center for Medium-Range Weather Forecasts (ECMWF) [36]

- **Initialization and Ensemble**. The operational IFS forecast is generated through Singular Vectors (SV) method: it creates a variety of initial conditions by adjusting certain parameters slightly, thus generating different starting points.

- **Coupling with ocean** is performed with NEMO3.4.1 with 1-degree resolution and 42 vertical pressure levels. Frequency of coupling is 3-hourly.

- **Coupling with sea ice** is not operational for this model's version (but it is in the newer generation, though the forecast start-date is much later than 2016). As a result, sea ice initial conditions are persisted up to day 15 and then relaxed to climatology up to day 45.

- **Coupling with wave model** is performed with ECMWF wave model with 0.5-degree resolution [69].

- **Coupling with land surface** is relatively more complex than the rest, and we refer readers to their documentation. Regardless, it is based on Land Data Assimilation System (LDAS) that combines heterogenous high-quality dataset from satellite to ground sensors, and integrated with the operational IFS model. The parameterization for land cover type is primarily based on MODIS collection 5 [57] and soil characteristics from the FAO dominant soil texture class [70].

- **Model grid** uses the Cubic Octohedral grid [71], where the Earth's surface is projected onto a cube. Then, the cube is further subdivided to form an octahedron, where the faces represent finer grid cells. This multi-scale gridding scheme allows for parallelization where processes at different scales could be solved simultaneously.

- **Large-scale dynamics** uses a mixture of Spectral and Semi-Lagrangian approach, similar to that used by CMA.

# E    Data-Driven Baseline Models

In this section, we describe in detail implementation and hyperparemeter selections of our data-driven models used as baselines to ChaosBench. Most of the choices are based on the original works that are adapted to weather and climate applications using similar input dataset. All training are performed using 2x NVIDIA A100 GPUs.

## E.1    Lagged Autoencoder (AE)

We implement lagged AE from [72] with 5 *encoder* blocks and 5 *decoder* block, with detailed specification in Table S4. Each encoder block is comprised of MAXPOOL2D $\circ$ (CONV2D $\rightarrow$ BATCHNORM2D $\rightarrow$ RELU $\rightarrow$ CONV2D $\rightarrow$ BATCHNORM2D $\rightarrow$ RELU). Similarly, the decoder block is comprised of CONVTRANSPOSE2D $\rightarrow$ BACTNORM2D $\rightarrow$ RELU) $\bigoplus$ (CONVTRANSPOSE2D $\rightarrow$ BACTNORM2D $\rightarrow$ SIGMOID) $\circ$ (CONV2D).

Table S4: Hyperparameters for Lagged AE

| Hyperparameters | Values |
|---|---|
| Channels | [64, 128, 256, 512, 1024] |
| Encoder Kernel | $3 \times 3$ |
| Decoder Kernel | $2 \times 2$ |
| Max Pooling Window | $2 \times 2$ |
| Batch Normalization | TRUE |
| Optimizer | ADAMW [73] |
| Learning Rate | COSINEANNEALING($10^{-2} \rightarrow 10^{-3}$) |
| Batch Size | 32 |
| Epochs | 500 |
| Tmax | 500 |

## E.2    ResNet

We adapt ResNet implementation from [42] using ResNet-50 as feature extractor and 5 *decoder* blocks, following specification in Table S5. Each decoder block is composed of CONVTRANSPOSE2D $\rightarrow$ BACTNORM2D $\rightarrow$ LEAKYRELU.

Table S5: Hyperparameters for ResNet

| Hyperparameters | Values |
|---|---|
| Backbone | RESNET-50 |
| Decoder Channels | [1024, 512, 256, 128, 64] |
| Decoder Activation | LEAKYRELU(0.15) |
| Optimizer | ADAMW |
| Learning Rate | COSINEANNEALING($10^{-2} \rightarrow 10^{-3}$) |
| Batch Size | 32 |
| Epochs | 500 |
| Tmax | 500 |

## E.3    UNet

We adapt UNet implementation from [22] using 5 *encoder* and 5 *decoder* blocks, with skip connections, following specification in Table S6. The composition of the encoder and decoder components are similar to those described for Lagged Autoencoder, with the addition of SKIP connection between each corresponding contracting-expansive path.

Table S6: Hyperparameters for UNet

| Hyperparameters | Values |
|---|---|
| Channels | [64, 128, 256, 512, 1024] |
| Activation | LEAKYRELU(0.15) |
| Encoder Kernel | $3 \times 3$ |
| Decoder Kernel | $2 \times 2$ |
| Max Pooling Window | $2 \times 2$ |
| Optimizer | ADAMW |
| Learning Rate | COSINEANNEALING($10^{-2} \rightarrow 10^{-3}$) |
| Batch Size | 32 |
| Epochs | 500 |
| Tmax | 500 |

## E.4 Fourier Neural Operator (FNO)

We adapt FNO implementation from [43], following specification in Table S7 and illustrated in S1. We implement the encoder-decoder structure, where we (1) first transform our input $\mathbf{X}_t$ by convolutional layers both in the Fourier (applying fast fourier transform; FFT) and physical domains, before we concatenate both (applying inverse FFT for the former convolved features), and apply non-linear GELU activation function [74]. We select only the first 4 main Fourier modes to make the number of trainable parameters comparable with the other data-driven baseline models. The (2) decoder block then applies deconvolutional operation to the latent features to generate output $Y_t$.

Table S7: Hyperparameters for FNO

| Hyperparameters | Values |
|---|---|
| Non-Spectral Channels | [64, 128, 256, 512, 1024] |
| Spectral Channel | [64, 128, 256, 512, 1024] |
| Activation | GELU |
| Fourier Modes | (4,4) |
| Optimizer | ADAMW |
| Learning Rate | COSINEANNEALING($10^{-2} \rightarrow 10^{-3}$) |
| Batch Size | 32 |
| Epochs | 500 |
| Tmax | 500 |

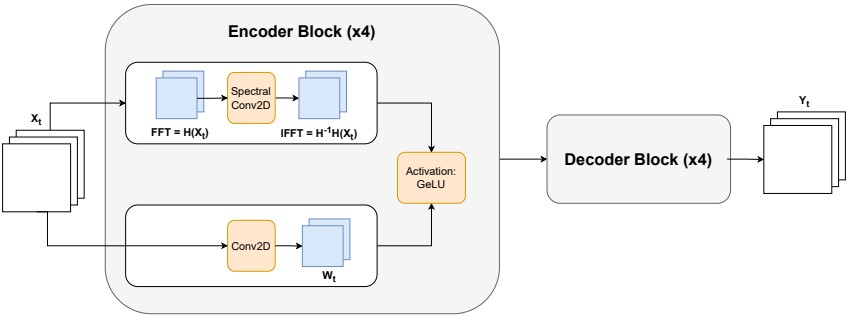

Figure S1: FNO architecture: (1) in the encoder block, we transform our input $X_t$ by convolutional layers both in the Fourier and physical domains, before we concatenate and apply non-linear GELU activation function. The (2) decoder block is then applying deconvolutional operation to the latent features to generate forecast $Y_t$.

### E.5 ClimaX

ClimaX is based on the ViT model [75] with variational positional embedding in variable-time space. We use ClimaX model as is described and implemented in the original paper and is pre-trained using CMIP6 [17]. We fine-tune the original pre-trained model given our training setup.

### E.6 PanguWeather, FourCastNetV2, GraphCast

We perform inference using their latest checkpoints using the API provided here: https://github.com/ecmwf-lab/ai-models.

For this work, we process the forecasts at biweekly temporal resolution. In the codebase, we provide the script for further flexibility, for instance:

```
# Process biweekly, 1.5-degree forecasts for the year 2022

## Panguweather
$ python scripts/process_sota.py --model_name panguweather --years
    2022

## Graphcast
$ python scripts/process_sota.py --model_name graphcast --years 2022

## FourCastNetV2
$ python scripts/process_sota.py --model_name fourcastnetv2 --years
    2022
```

# F Evaluation Metrics

## F.1 Deterministic Metrics

We describe in detail the four primary vision-based metrics used for this benchmark, including RMSE, Bias, ACC, and MS-SSIM.

### F.1.1 Root Mean-Squared Error (RMSE)

As described in the main text, we apply latitude-adjustment to RMSE computation.

$$\mathcal{M}_{RMSE} = \sqrt{\frac{1}{|\boldsymbol{\theta}||\boldsymbol{\gamma}|} \sum_{i=1}^{|\boldsymbol{\theta}|} \sum_{j=1}^{|\boldsymbol{\gamma}|} w(\theta_i)(\hat{\mathbf{Y}}_{i,j} - \mathbf{Y}_{i,j})^2} \tag{S1}$$

### F.1.2 Bias

Similarly, we apply latitude-adjustment to Bias computation.

$$\mathcal{M}_{Bias} = \frac{1}{|\boldsymbol{\theta}||\boldsymbol{\gamma}|} \sum_{i=1}^{|\boldsymbol{\theta}|} \sum_{j=1}^{|\boldsymbol{\gamma}|} w(\theta_i)(\hat{\mathbf{Y}}_{i,j} - \mathbf{Y}_{i,j}) \tag{S2}$$

### F.1.3 Anomaly Correlation Coefficient (ACC)

We remove the indexing for a more compact representation where the summation is performed over each grid cell $(i, j)$. The predicted and observed anomalies at each grid-cell are denoted by $A_{\hat{\mathbf{Y}}_{i,j}} = \hat{\mathbf{Y}}_{i,j} - C$ and $A_{\mathbf{Y}_{i,j}} = \mathbf{Y}_{i,j} - C$, where $C$ is the observational climatology. We apply latitude-adjustment to ACC computation.

$$\mathcal{M}_{ACC} = \frac{\sum w(\theta)[A_{\hat{\mathbf{Y}}} \cdot A_{\mathbf{Y}}]}{\sqrt{\sum w(\theta)A_{\hat{\mathbf{Y}}}^2 \sum w(\theta)A_{\mathbf{Y}}^2}} \tag{S3}$$

### F.1.4 Multi-scale Structural Similarity Index Measure (MS-SSIM)

Let $\mathbf{Y}$ and $\hat{\mathbf{Y}}$ be two images to be compared, and let $\mu_{\mathbf{Y}}$, $\sigma_{\mathbf{Y}}^2$ and $\sigma_{\mathbf{Y}\hat{\mathbf{Y}}}$ be the mean of $\mathbf{Y}$, the variance of $\mathbf{Y}$, and the covariance of $\mathbf{Y}$ and $\hat{\mathbf{Y}}$, respectively. The luminance, contrast and structure comparison measures are defined as follows:

$$l(\mathbf{Y}, \hat{\mathbf{Y}}) = \frac{2\mu_{\mathbf{Y}}\mu_{\hat{\mathbf{Y}}} + C_1}{\mu_{\mathbf{Y}}^2 + \mu_{\hat{\mathbf{Y}}}^2 + C_1}, \tag{S4}$$

$$c(\mathbf{Y}, \hat{\mathbf{Y}}) = \frac{2\sigma_{\mathbf{Y}}\sigma_{\hat{\mathbf{Y}}} + C_2}{\sigma_{\mathbf{Y}}^2 + \sigma_{\hat{\mathbf{Y}}}^2 + C_2}, \tag{S5}$$

$$s(\mathbf{Y}, \hat{\mathbf{Y}}) = \frac{\sigma_{\mathbf{Y}\hat{\mathbf{Y}}} + C_3}{\sigma_{\mathbf{Y}}\sigma_{\hat{\mathbf{Y}}} + C_3}, \tag{S6}$$

where $C_1$, $C_2$ and $C_3$ are constants given by

$$C_1 = (K_1 L)^2, C_2 = (K_2 L)^2, \text{ and } C_3 = C_2/2. \tag{S7}$$

$L = 255$ is the dynamic range of the gray scale images, and $K_1 \ll 1$ and $K_2 \ll 1$ are two small constants. To compute the MS-SSIM metric across multiple scales, the images are successively low-pass filtered and down-sampled by a factor of 2. We index the original image as scale 1, and the desired highest scale as scale $M$. At each scale, the contrast comparison and structure comparison

are computed and denoted as $c_j(\mathbf{Y}, \hat{\mathbf{Y}})$ and $s_j(\mathbf{Y}, \hat{\mathbf{Y}})$ respectively. The luminance comparison is only calculated at the last scale $M$, denoted by $l_M(\mathbf{Y}, \hat{\mathbf{Y}})$. Then, the MS-SSIM metric is defined by

$$\mathcal{M}_{MS-SSIM} = [l_M(\mathbf{Y}, \hat{\mathbf{Y}})]^{\alpha_M} \cdot \prod_{j=1}^{M} [c_j(\mathbf{Y}, \hat{\mathbf{Y}})]^{\beta_j} [s_j(\mathbf{Y}, \hat{\mathbf{Y}})]^{\gamma_j} \tag{S8}$$

where $\alpha_M$, $\beta_j$ and $\gamma_j$ are parameters. We use the same set of parameters as in [37]: $K_1 = 0.01$, $K_2 = 0.03$, $M = 5$, $\alpha_5 = \beta_5 = \gamma_5 = 0.1333$, $\beta_4 = \gamma_4 = 0.2363$, $\beta_3 = \gamma_3 = 0.3001$, $\beta_2 = \gamma_2 = 0.2856$, $\beta_1 = \gamma = 0.0448$. The predicted and ground truth images of physical variables are re-scaled to 0-255 prior to the calculation of their MS-SSIM values.

## F.2 Physics-Based Metrics

In this section, we describe in detail the definition and implementation of our physics-based metrics, including PYTORCH psuedocode implementation.

Let $\mathbf{Y}$ be a 2D image of size $h \times w$ for a physical variables at a specific time, variable, and level. Let $f(x, y)$ be the intensity of the pixel at position $(x, y)$. First, we compute the 2D Fourier transform of the image by

$$F(k_x, k_y) = \sum_{x=0}^{w-1} \sum_{y=0}^{h-1} f(x, y) \cdot e^{-2\pi i(k_x x/w + k_y y/h)}, \tag{S9}$$

where $k_x$ and $k_y$ correspond to the wavenumber components in the horizontal and vertical directions, respectively, and $i$ is the imaginary unit. The power at each wavenumber component $(k_x, k_y)$ is given by the square of the magnitude spectrum of $F(k_x, k_y)$, that is,

$$S(k_x, k_y) = |F(k_x, k_y)|^2 = \texttt{Re}[F(k_x, k_y)]^2 + \texttt{Im}[F(k_x, k_y)]^2. \tag{S10}$$

The scalar wavenumber is defined as:

$$k = \sqrt{k_x^2 + k_y^2}, \tag{S11}$$

which represents the magnitude of the spatial frequency vector, indicating how rapidly features change spatially regardless of direction. Then, the energy distribution at a spatial frequency corresponding to k is defined as

$$S(k) = \sum_{(k_x, k_y): \sqrt{k_x^2 + k_y^2} = k} S(k_x, k_y). \tag{S12}$$

Given the spatial energy frequency distribution for observations $E(k)$ and predictions $\hat{S}(k)$ , we perform normalization for each over $\mathbf{K}_q$, the set of wavenumbers corresponding to high-frequency components of energy distribution, as defined in Equation S13. This is to ensure that the sum of the component sums up to 1 which exhibits pdf-like property.

$$S'(k) = \frac{S(k)}{\sum_{k \in \mathbf{K}_q} S(k)}, \quad \hat{S}'(k) = \frac{\hat{S}(k)}{\sum_{k \in \mathbf{K}_q} \hat{S}(k)}, \quad k \in \mathbf{K}_q \tag{S13}$$

```python
import torch
import torch.nn as nn

class SpectralDiv(nn.Module):
    """
    Compute Spectral divergence given the top-k percentile wavenumber
        (higher k means higher frequency)
    """
    def __init__(
        self,
        percentile=0.9,
        input_shape=(121,240)
    ):
        super(SpectralDiv, self).__init__()

        self.percentile = percentile

        # Compute the discrete Fourier Transform sample frequencies
        #     for a signal of size
        nx, ny = input_shape
        kx = torch.fft.fftfreq(nx) * nx
        ky = torch.fft.fftfreq(ny) * ny
        kx, ky = torch.meshgrid(kx, ky)

        # Construct discretized k-bins
        self.k = specify_k_bins(...)

        # Get k-percentile index
        self.k_percentile_idx = int(len(self.k) * self.percentile)

    def forward(self, predictions, targets):

        # Preprocess data, including handling of missing values, etc
        predictions = preprocess_data(...)
        targets = preprocess_data(...)

        # Compute along mini-batch
        predictions, targets = torch.nanmean(predictions, dim=0), \
            torch.nanmean(targets, dim=0)

        # Transform prediction and targets onto the Fourier space and
        #     compute the power
        predictions_power = torch.fft.fft2(predictions)
        predictions_power = torch.abs(predictions_power)**2

        targets_power = torch.fft.fft2(targets)
        targets_power = torch.abs(targets_power)**2

        # Normalize as pdf
        predictions_Sk = predictions_power / torch.nansum(
            predictions_power)
        targets_Sk = targets_power / torch.nansum(targets_power)

        # Compute spectral Sk divergence
        div = torch.nansum(targets_Sk * torch.log(torch.clamp(
            targets_Sk / predictions_Sk, min=1e-9)))

        return div
```

Listing S1: Psuedocode for computing SpecDiv using PYTORCH

```python
import torch
import torch.nn as nn

class SpectralRes(nn.Module):
    """
    Compute Spectral residual given the top-k percentile wavenumber (
        higher k means higher frequency)
    """
    def __init__(
        self,
        percentile=0.9,
        input_shape=(121,240)
    ):
        super(SpectralRes, self).__init__()

        self.percentile = percentile

        # Compute the discrete Fourier Transform sample frequencies
        #     for a signal of size
        nx, ny = input_shape
        kx = torch.fft.fftfreq(nx) * nx
        ky = torch.fft.fftfreq(ny) * ny
        kx, ky = torch.meshgrid(kx, ky)

        # Construct discretized k-bins
        self.k = specify_k_bins(...)

        # Get k-percentile index
        self.k_percentile_idx = int(len(self.k) * self.percentile)

    def forward(self, predictions, targets):

        # Preprocess data, including handling of missing values, etc
        predictions = preprocess_data(...)
        targets = preprocess_data(...)

        # Compute along mini-batch
        predictions, targets = torch.nanmean(predictions, dim=0),
            torch.nanmean(targets, dim=0)

        # Transform prediction and targets onto the Fourier space and
        #     compute the power
        predictions_power = torch.fft.fft2(predictions)
        predictions_power = torch.abs(predictions_power)**2

        targets_power = torch.fft.fft2(targets)
        targets_power = torch.abs(targets_power)**2

        # Normalize as pdf
        predictions_Sk = predictions_power / torch.nansum(
            predictions_power)
        targets_Sk = targets_power / torch.nansum(targets_power)

        # Compute spectral Sk residual
        res = torch.sqrt(torch.nanmean(torch.square(predictions_Sk -
            targets_Sk)))

        return res
```

Listing S2: Psuedocode for computing SpecRes using PYTORCH

### F.3 Probabilistic Metrics

Here, we broadly define $n \in N$ as an ensemble member, and $N \in \mathbb{R}$ the total number of ensemble members.

#### F.3.1 Deterministic Extension

This includes the ensemble version of deterministic and physics-based metrics, including RMSE, Bias, ACC, MS-SSIM, SpecDiv, and SpecRes.

$$\mathcal{M}_{RMSE}^{ens} = \frac{1}{N} \sum_{n=1}^{N} \mathcal{M}_{RMSE}^{n} \tag{S14}$$

$$\mathcal{M}_{Bias}^{ens} = \frac{1}{N} \sum_{n=1}^{N} \mathcal{M}_{Bias}^{n} \tag{S15}$$

$$\mathcal{M}_{ACC}^{ens} = \frac{1}{N} \sum_{n=1}^{N} \mathcal{M}_{ACC}^{n} \tag{S16}$$

$$\mathcal{M}_{MS-SSIM}^{ens} = \frac{1}{N} \sum_{n=1}^{N} \mathcal{M}_{MS-SSIM}^{n} \tag{S17}$$

$$\mathcal{M}_{SpecDiv}^{ens} = \frac{1}{N} \sum_{n=1}^{N} \mathcal{M}_{SpecDiv}^{n} \tag{S18}$$

$$\mathcal{M}_{SpecRes}^{ens} = \frac{1}{N} \sum_{n=1}^{N} \mathcal{M}_{SpecRes}^{n} \tag{S19}$$

#### F.3.2 CRPS

CRPS measures the accuracy of probabilistic forecasts by integrating the square of the difference between the cumulative distribution function (CDF) of the forecast and the CDF of the observed data over all possible outcomes. It can be thought of as probabilistic MAE, where a smaller value is desirable and a deterministic forecast reduces to MAE. We first apply latitude-adjustments for the forecasts and target fields.

$$\mathcal{M}_{CRPS}(F, x) = \int_{-\infty}^{\infty} (F(y) - H(y - x))^2 \, dy \tag{S20}$$

where $F(y)$ is the CDF of the forecast, $H(y - x)$ is the Heaviside step function at the observed value $x$, and $y$ ranges over all possible outcomes.

#### F.3.3 CRPSS

CRPSS measures the skillfulness of an ensemble forecasts, with positive being skillful, zero unskilled, and negative being worse than baseline climatology.

$$\mathcal{M}_{CRPSS} = 1 - \frac{CRPS_{\text{forecast}}}{CRPS_{\text{climatology}}} \tag{S21}$$

### F.3.4 Spread

We apply latitude-adjusted spread of the ensemble members, and std is the standard deviation operator.

$$\mathcal{M}_{Spread} = \frac{1}{|\boldsymbol{\theta}||\boldsymbol{\gamma}|} \sum_{i=1}^{|\boldsymbol{\theta}|} \sum_{j=1}^{|\boldsymbol{\gamma}|} \texttt{std}\left(\{w(\theta_i)\hat{\mathbf{Y}}_{i,j}^n\}_{n=1}^N\right) \tag{S22}$$

### F.3.5 Spread/Skill Ratio (SSR)

We use ensemble RMSE as the skill in the SSR computation.

$$\mathcal{M}_{SSR} = \frac{\mathcal{M}_{Spread}}{\mathcal{M}_{RMSE}^{ens}} \tag{S23}$$

# G  Extended Results

We provide extended results accompanying the main text.

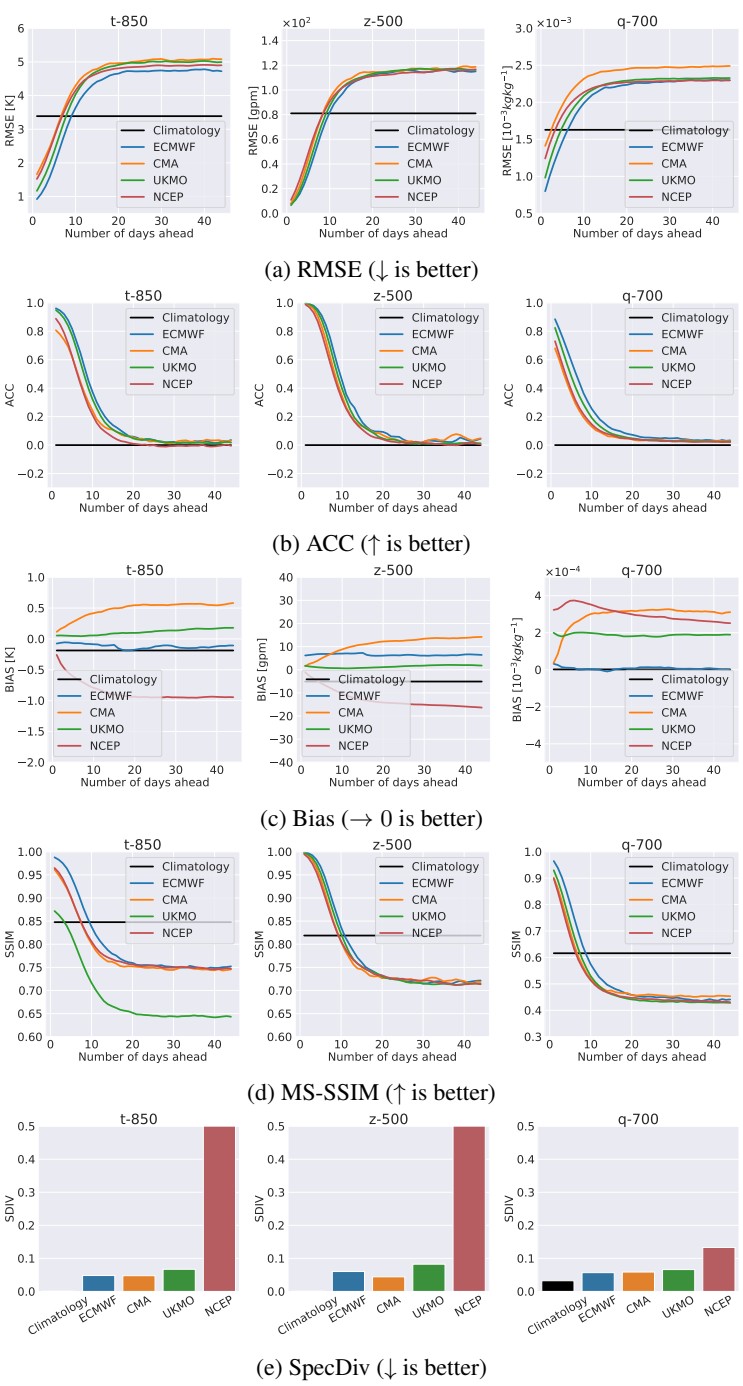

(a) RMSE ($\downarrow$ is better)

(b) ACC ($\uparrow$ is better)

(c) Bias ($\rightarrow 0$ is better)

(d) MS-SSIM ($\uparrow$ is better)

(e) SpecDiv ($\downarrow$ is better)

Figure S2: Evaluation results between baseline climatology (black line) and physics-based control (deterministic) forecasts. At longer forecasting horizon, most physics-based deterministic forecasts perform worse than climatology while maintaining structures as evidenced from their low SpecDiv (barring NCEP).

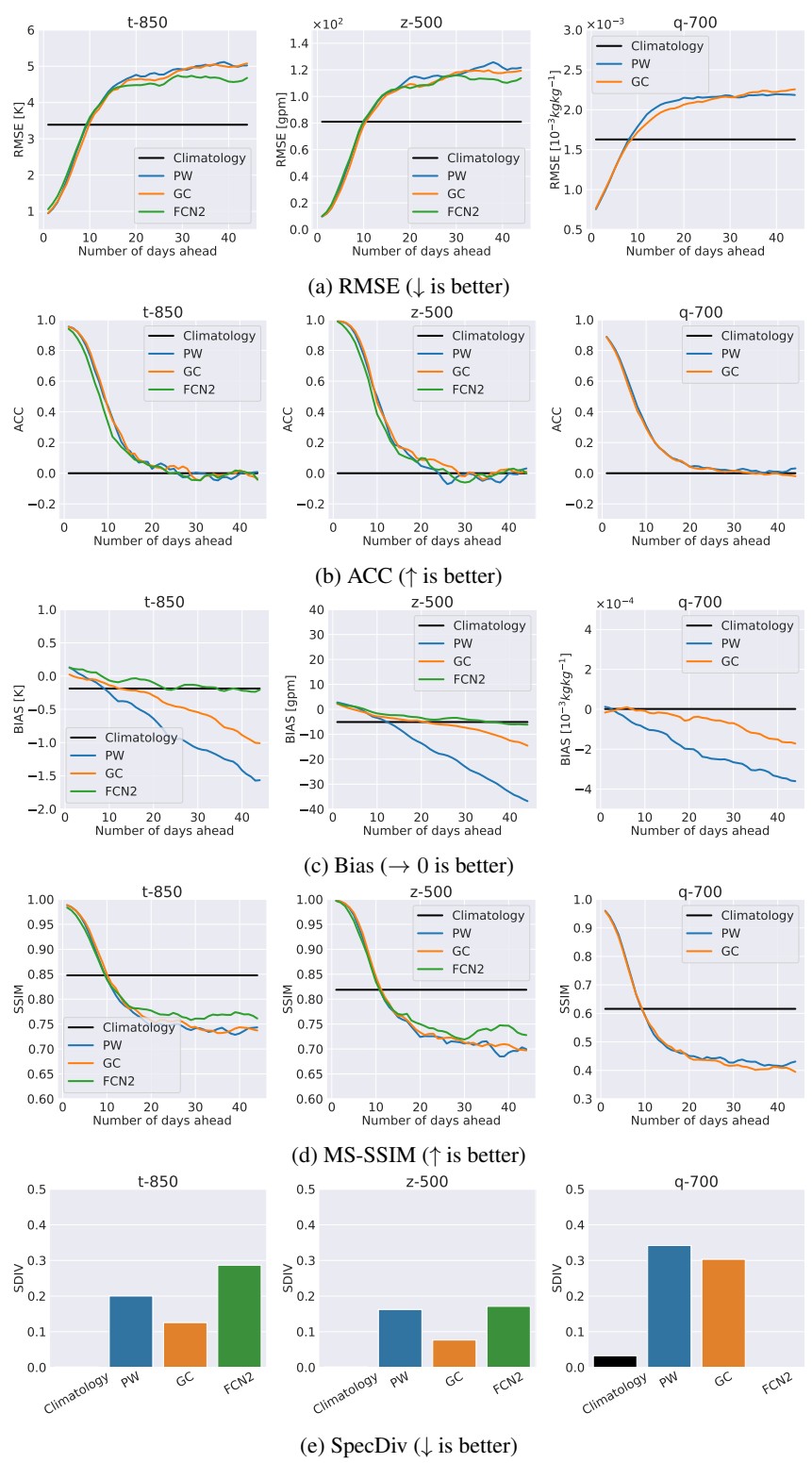

(a) RMSE (↓ is better)

(b) ACC (↑ is better)

(c) Bias (→ 0 is better)

(d) MS-SSIM (↑ is better)

(e) SpecDiv (↓ is better)

Figure S3: Evaluation results between baseline climatology (black line) and data-driven models including PanguWeather (PW), FourCastNetV2 (FCN2), and GraphCast (GC). Overall, we observe that data-driven models perform significantly worse than climatology on S2S timescale. They also perform poorly on physics-based metrics indicating the lack of predictive power on multi-scale structures. Note: FCN2 lacks q-700 and climatology naturally has low SpecDiv (direct observations).

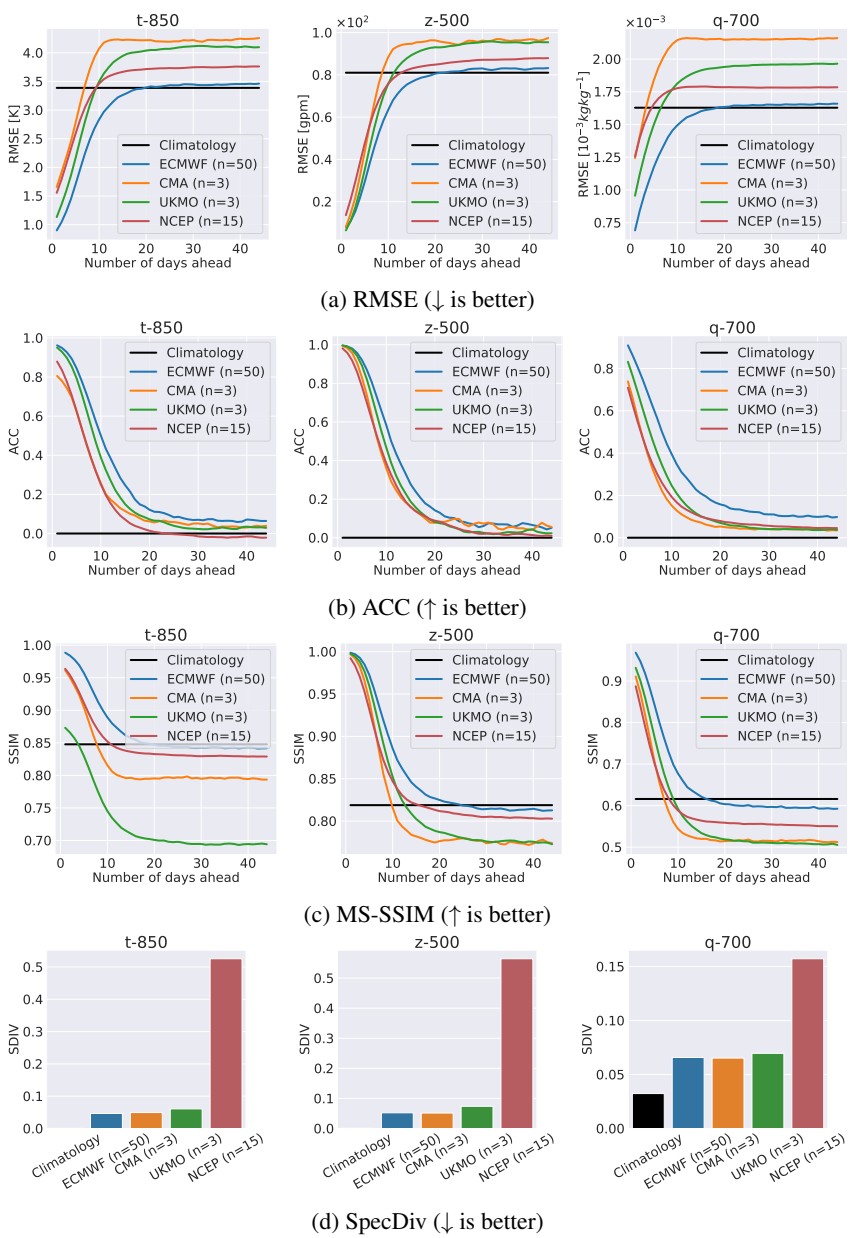

(a) RMSE (↓ is better)

(b) ACC (↑ is better)

(c) MS-SSIM (↑ is better)

(d) SpecDiv (↓ is better)

Figure S4: Evaluation results between baseline climatology (black line) and physics-based ensembles from ECMWF, CMA, UKMO, NCEP. Overall, we observe that ensemble forecasts perform better than their deterministic counterparts.

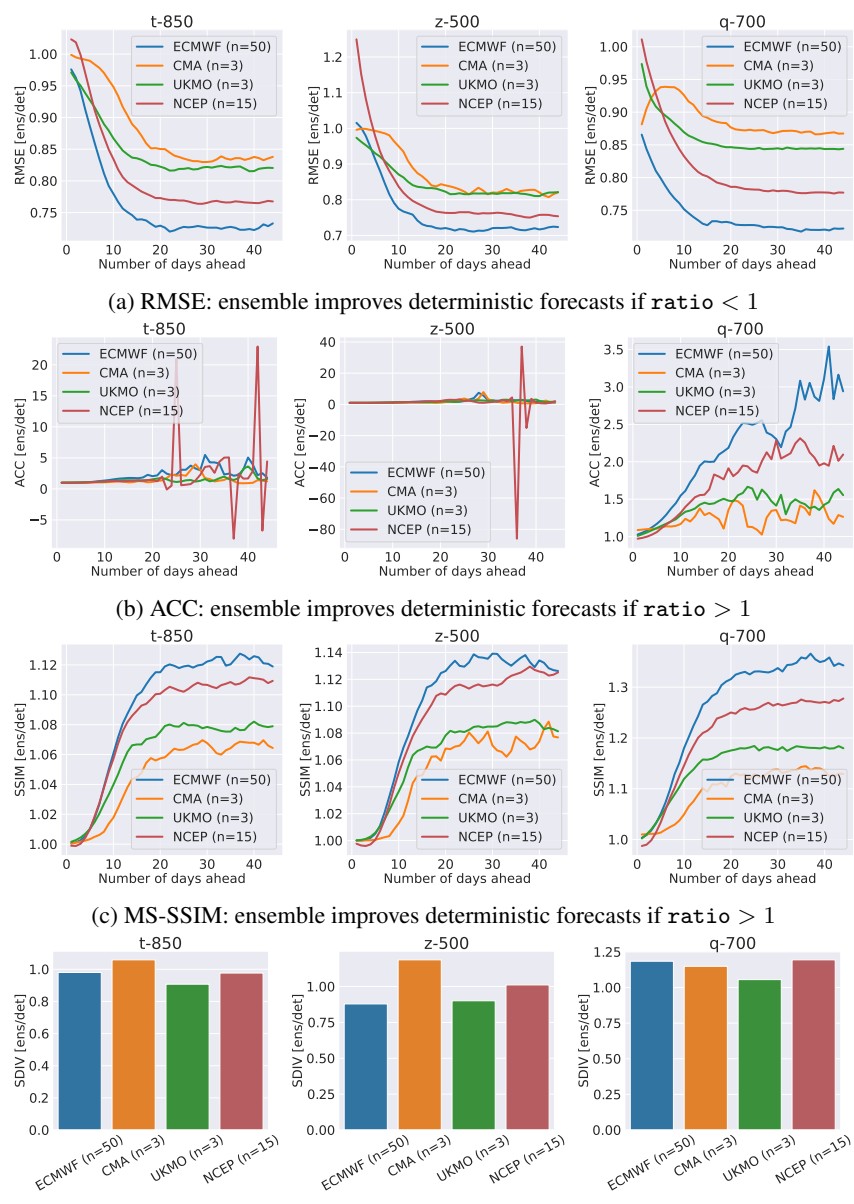

(a) RMSE: ensemble improves deterministic forecasts if `ratio` $< 1$

(b) ACC: ensemble improves deterministic forecasts if `ratio` $> 1$

(c) MS-SSIM: ensemble improves deterministic forecasts if `ratio` $> 1$

(d) SpecDiv: ensemble improves deterministic forecasts if `ratio` $< 1$

Figure S5: Metrics ratio e.g., $\text{RMSE}_{ens}/\text{RMSE}_{det}$ between ensemble and deterministic forecasts, where the former improves the latter by accounting for IC uncertainty that can lead to long-range instability and trajectory divergences. Note: $n$ represents the number of ensemble members. The ratio for `ACC` fluctuates as the scalar value approaches 0.

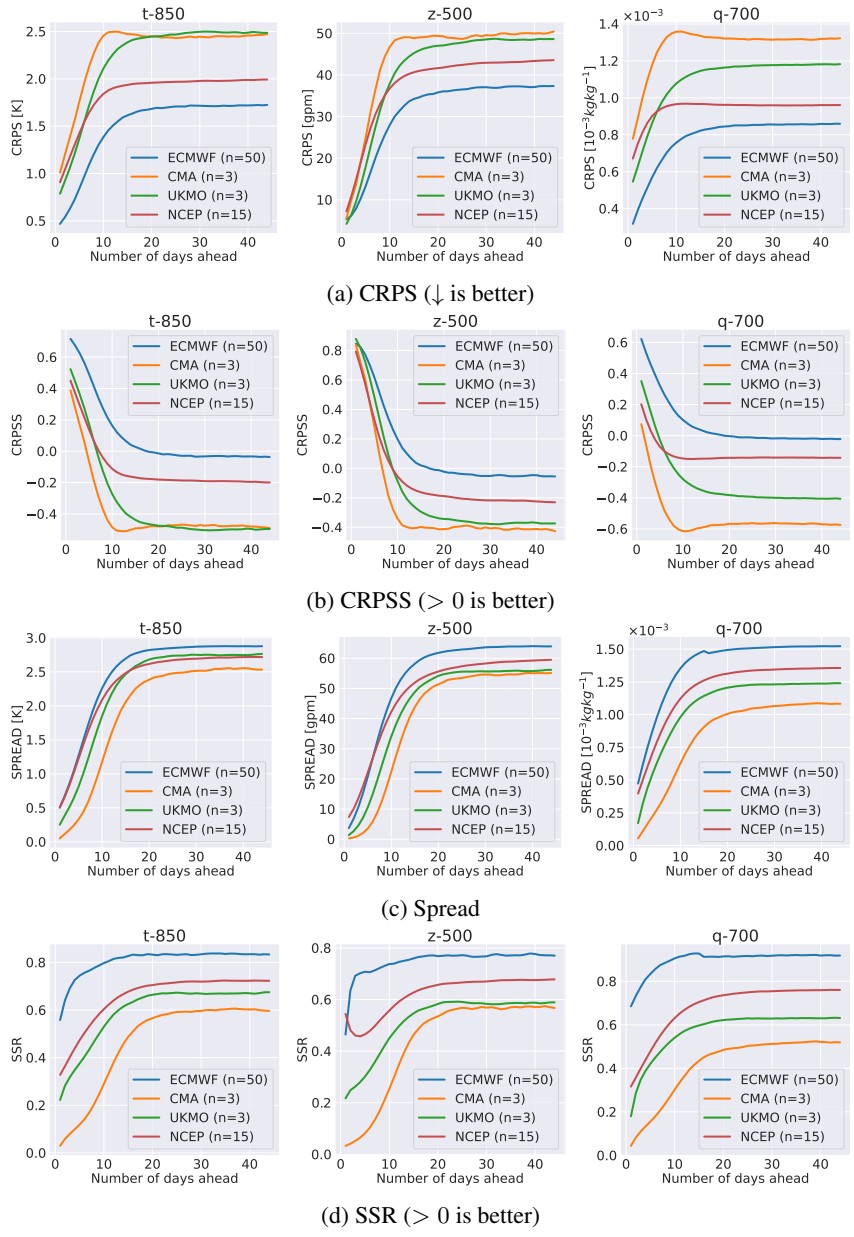

Figure S6: Probabilistic evaluation on ensemble forecasts indicating current skill limits of 15 days. Note: $n$ represents the number of ensemble members.

## G.1 Effects of Different Autoregressive Training Steps; `lead_time`

We showcased more results for autoregressive training strategy. In this case, we performed autoregressive training using either 1 or 5 iterative steps (`n_step`; $s$). As illustrated in Figure S7, we observe that incorporating temporal information improve the vision-based metrics even at longer forecasting timesteps, with lower RMSE, higher MS-SSIM. However, the converse trend is true incorporating temporal context makes S2S forecast worse off in some physics-based scores. The modified loss function for training a model with multiple autoregressive steps is:

$$\mathcal{L} = \frac{1}{|S|} \sum_{i=1}^{s} \mathcal{L}(\hat{\mathbf{Y}}_{\mathbf{t+s_i}} \mathbf{Y}_{\mathbf{t+s_i}}), \forall s_i \in S \tag{S24}$$

Here $S = \{1, \cdots, s\}$ and $s \in \mathbb{N}^+$ is the autoregressive steps. For this work, we set $s = 5$.

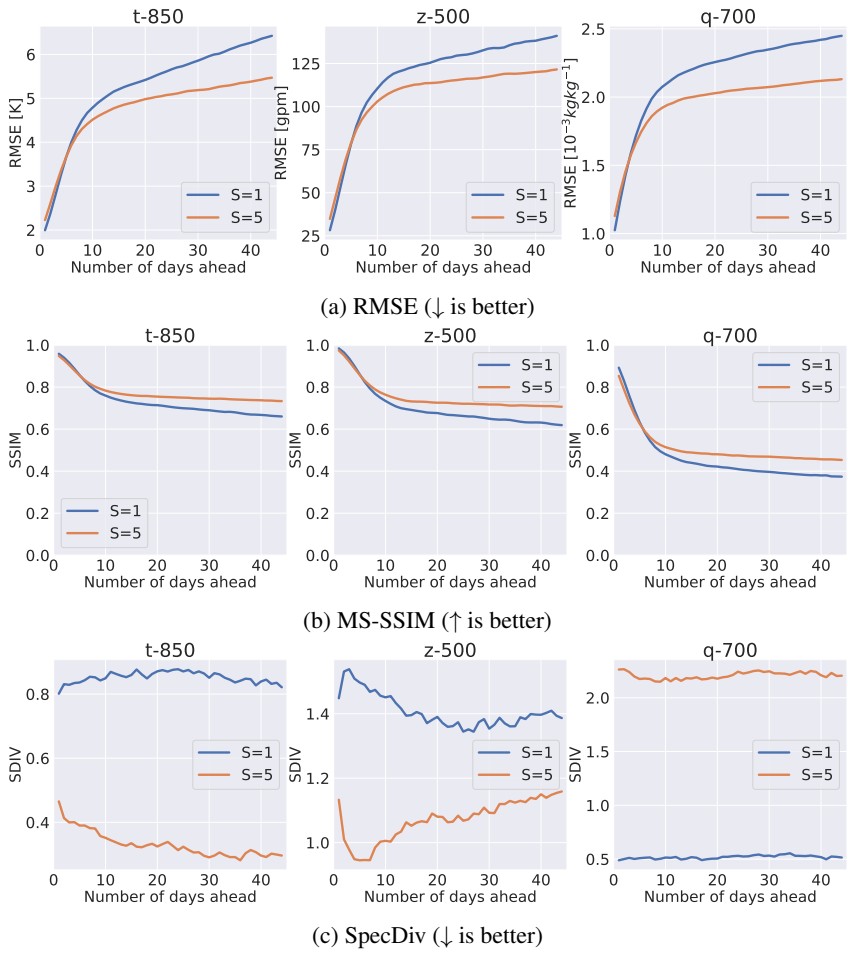

(a) RMSE ($\downarrow$ is better)

(b) MS-SSIM ($\uparrow$ is better)

(c) SpecDiv ($\downarrow$ is better)

Figure S7: Ablation results for incorporating temporal information in an autoregressive scheme for long-range forecast using UNet models. The x-axis represents the number of forecasting days for t-850, z-500, q-700 representative tasks. Blue and orange lines illustrate autoregressive scheme with $s = 1$ and $s = 5$ respectively. Overall we observe that incorporating temporal information improve the vision-based metrics even at longer forecasting timesteps. However, the converse trend is true where incorporating temporal context makes S2S forecast worse off in some physics-based scores.

## G.2 Effects of Subset Optimization; `headline_vars`

In many cases, we seek to train data-driven models so that they are able to perform well on *all* states by optimizing for the full state of the next forecasting timestep $t + 1$, that is,

$$\phi^* = \underset{\phi}{\operatorname{argmin}} \, \mathcal{L}(\hat{\mathbf{Y}}_{t+1}, \mathbf{Y}_{t+1})$$

where $\mathcal{L}$ is any loss function. This task is especially useful for building emulators that act as surrogates for the more expensive physics-based NWP models [17].

Although the first task is useful for learning the full complex interaction between variables, it is relatively difficult due to the intrinsic high-dimensionality of the data. As a result, we introduce a second task that allows for the optimization on a subset of variables of interest ($\mathbf{Y}' \in \mathbf{Y}$):

$$\phi^* = \underset{\phi}{\operatorname{argmin}} \, \mathcal{L}(\hat{\mathbf{Y}}'_{t+1}, \mathbf{Y}'_{t+1})$$

Here $\mathbf{Y}'_{t+1} = \{$t-850, z-500, q-700$\}$, and we train them using 5 autoregressive steps i.e., `n_step` = 5.

Table S8: Long-range forecasting ($\Delta t = 44$) results on select metrics and target variables between physics-based and data-driven models. Results are for Task 1 (full) and Task 2 (sparse). *(\*) Baseline model that uses privileged information (observations) to make prediction.*

| Models | RMSE ↓ | | | MS-SSIM ↑ | | | SpecDiv ↓ | | |
|---|---|---|---|---|---|---|---|---|---|
| | T850 $(K)$ | Z500 $(gpm)$ | Q700 $(\times 10^{-3})$ | T850 | Z500 | Q700 | T850 | Z500 | Q700 |
| Climatology* | **3.39** | **81.0** | **1.62** | **0.85** | **0.82** | **0.62** | **0.01** | **0.01** | **0.03** |
| Persistence* | 5.88 | 127.8 | 2.47 | 0.71 | 0.69 | 0.41 | 0.02 | 0.03 | 0.05 |
| UKMO | 5.00 | 116.2 | 2.32 | 0.64 | 0.71 | 0.43 | 0.06 | 0.09 | 0.07 |
| NCEP | 4.90 | 116.7 | 2.30 | 0.75 | 0.71 | 0.43 | 0.53 | 0.55 | 0.10 |
| CMA | 5.08 | 118.7 | 2.49 | 0.75 | 0.72 | 0.45 | 0.05 | 0.04 | 0.06 |
| ECMWF | 4.72 | 115.1 | 2.30 | 0.75 | 0.72 | 0.44 | 0.06 | 0.07 | 0.06 |
| | Task 1: Full Dynamics Prediction | | | | | | | | |
| Lagged AE | 5.55 | 122.4 | 2.03 | 0.74 | 0.71 | 0.47 | 0.18 | 2.44 | 0.21 |
| ResNet | 5.67 | 125.3 | 2.07 | 0.73 | 0.70 | 0.47 | 0.21 | 0.37 | 0.26 |
| UNet | 5.47 | 121.5 | 2.13 | 0.73 | 0.71 | 0.45 | 0.30 | 1.16 | 2.20 |
| FNO | **5.06** | **112.5** | **1.95** | **0.75** | **0.73** | **0.51** | **0.18** | **0.11** | **0.10** |
| | Task 2: Sparse Dynamics Prediction | | | | | | | | |
| Lagged AE | 5.39 | 119.0 | 2.12 | 0.75 | 0.73 | 0.48 | 0.52 | 1.41 | 0.29 |
| ResNet | 5.80 | 124.1 | 2.18 | 0.74 | 0.72 | 0.46 | 0.33 | 1.22 | 0.09 |
| UNet | 5.57 | 120.2 | 2.18 | 0.74 | 0.71 | 0.45 | 1.20 | 1.08 | **0.07** |
| FNO | **4.73** | **101.8** | **1.91** | **0.79** | **0.76** | **0.52** | **0.18** | **0.23** | 0.21 |

Overall, we find models that attempt to preserve spectral structures (e.g., FNO) perform better on all metrics, deterministic and physics-based. Also, Task 2 (sparse) appears to be easier than Task 1 (full). Nonetheless, they are still performing worse than climatology.

## G.3 Effects of Ensemble Forecasts

This section provides additional results for data-driven ensemble approach, and follow similar evaluation process as the physics-based counterpart.

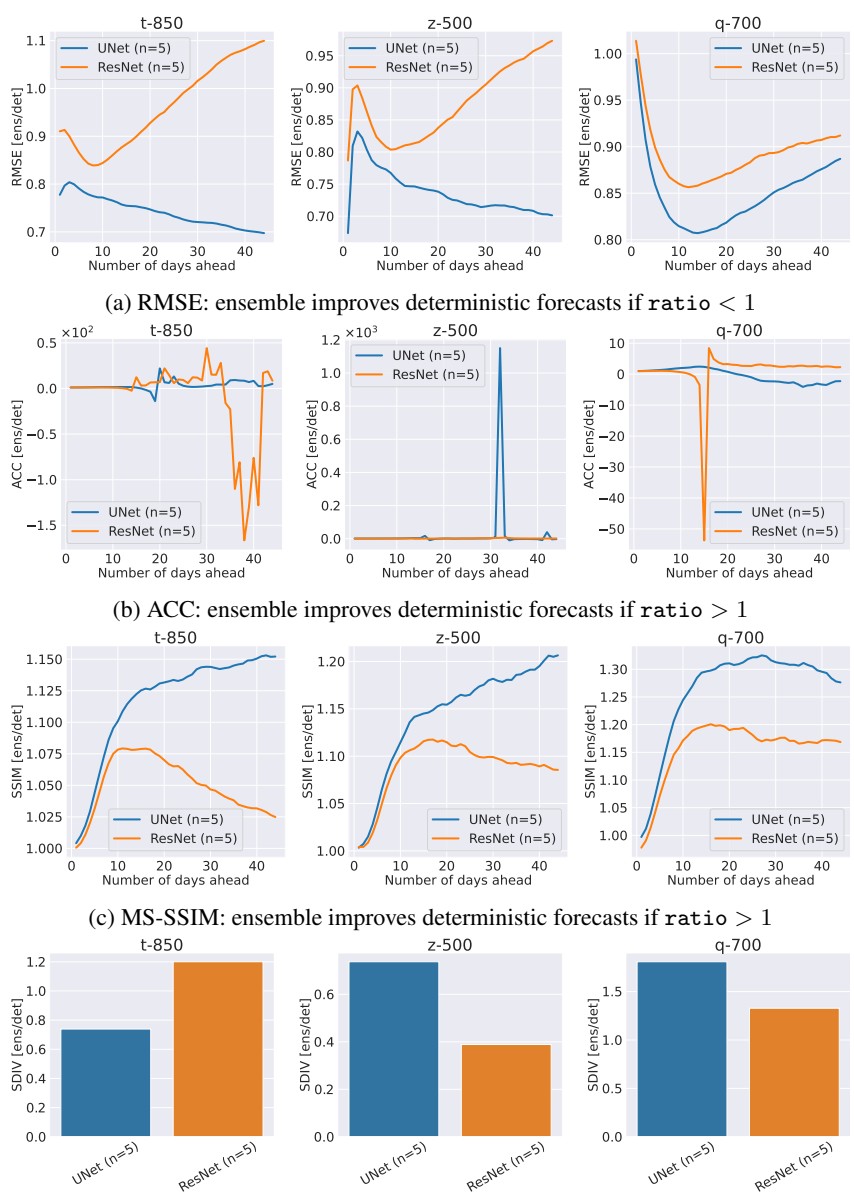

(a) RMSE: ensemble improves deterministic forecasts if `ratio` $< 1$

(b) ACC: ensemble improves deterministic forecasts if `ratio` $> 1$

(c) MS-SSIM: ensemble improves deterministic forecasts if `ratio` $> 1$

(d) SpecDiv: ensemble improves deterministic forecasts if `ratio` $< 1$

Figure S8: Metrics ratio e.g., $\text{RMSE}_{ens}/\text{RMSE}_{det}$ between ensemble and deterministic forecasts, where the former improves the latter by accounting for IC uncertainty that can lead to long-range instability and trajectory divergences. Note: $n$ represents the number of ensemble members.

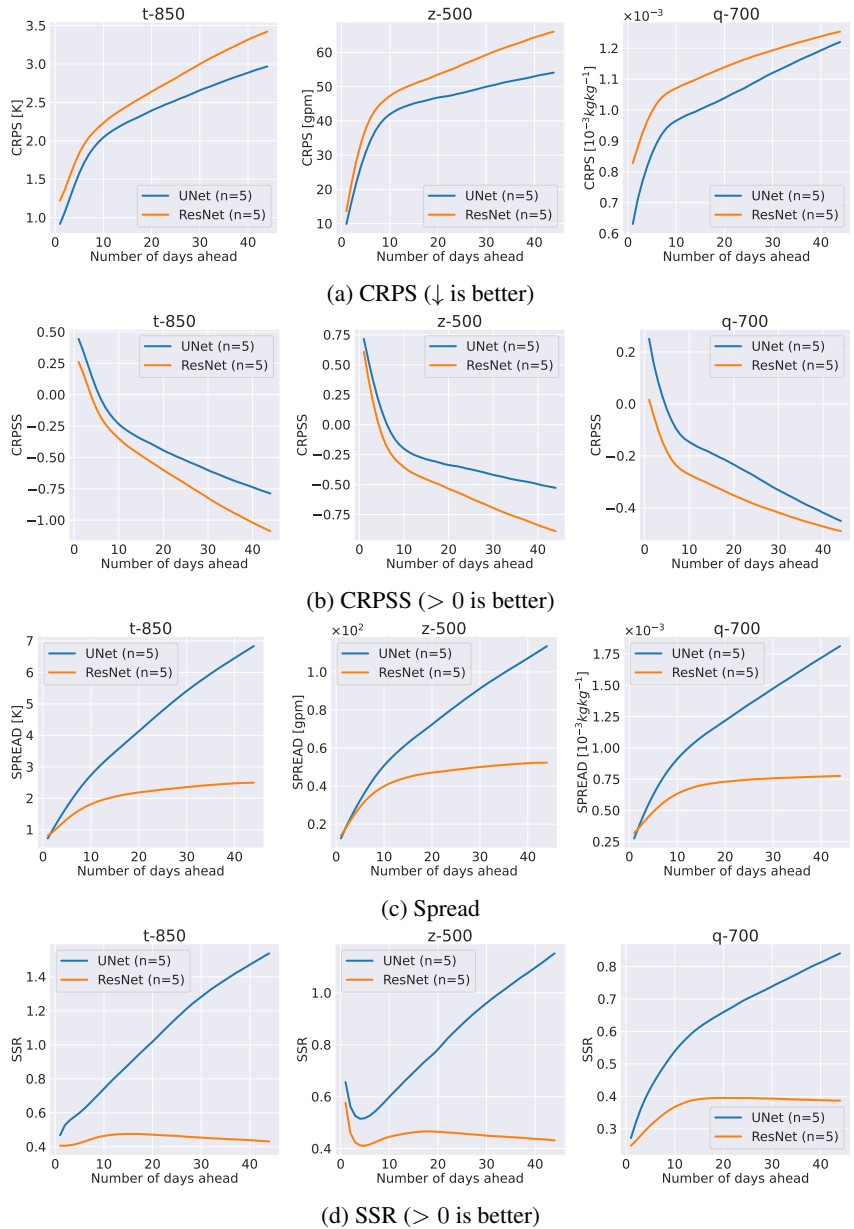

(a) CRPS (↓ is better)

(b) CRPSS (> 0 is better)

(c) Spread

(d) SSR (> 0 is better)

Figure S9: Probabilistic evaluation on ensemble forecasts. Note: $n$ represents the number of ensemble members.

### G.4 Power Spectra

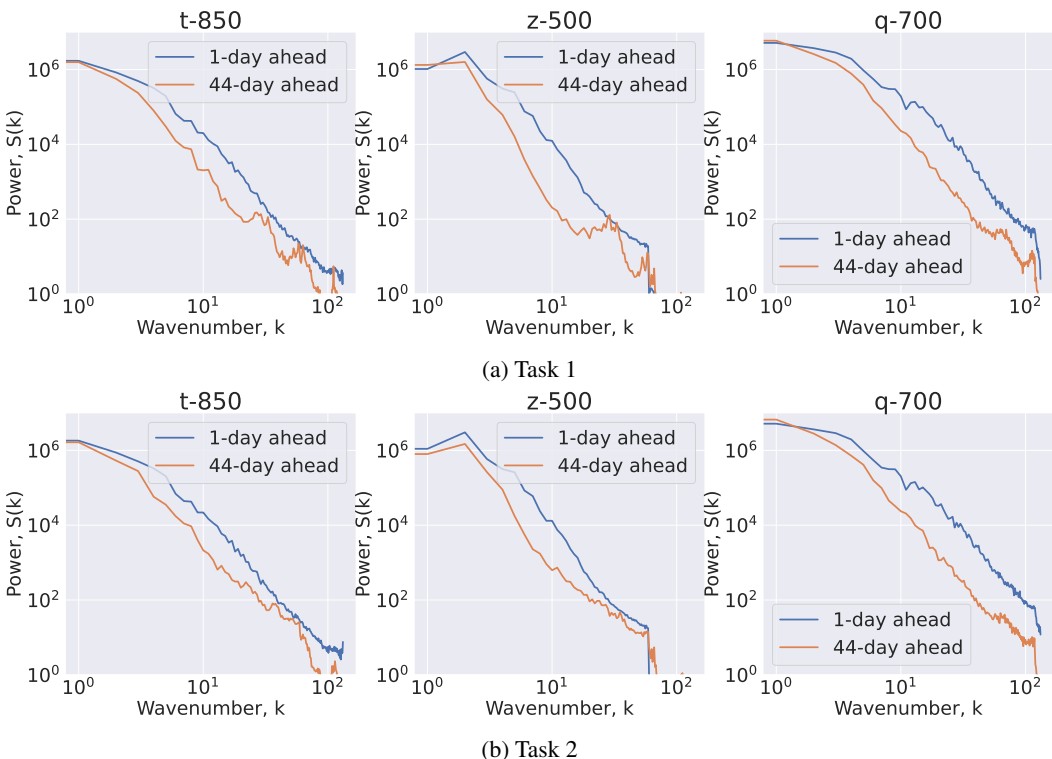

(a) Task 1

(b) Task 2

Figure S10: Power spectra for ViT/ClimaX demonstrating energy decay/divergence especially for high $k$ as lead time grows.

## G.5 Qualitative Evaluation

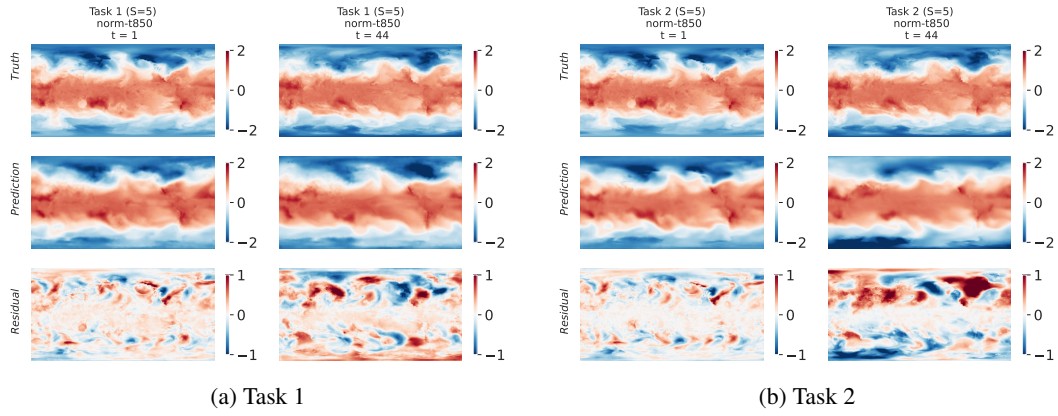

(a) Task 1          (b) Task 2

Figure S11: Normalized t@850-hpa qualitative results for UNet-autoregressive (S=5).

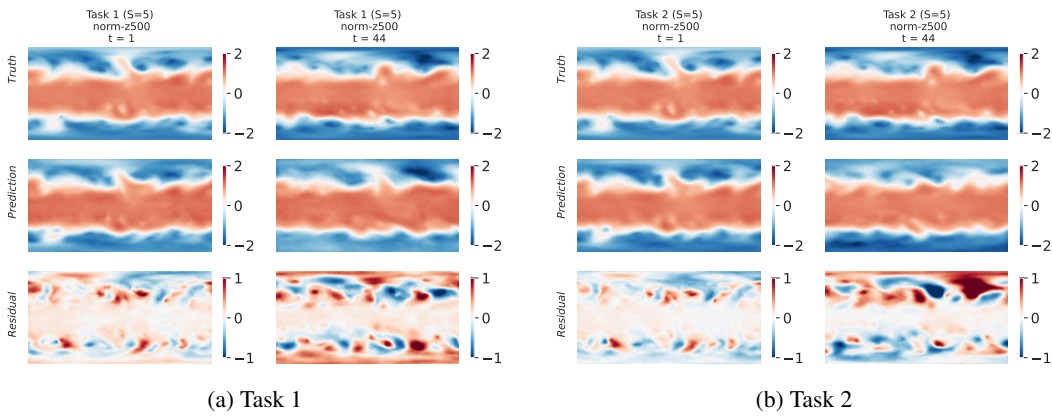

(a) Task 1          (b) Task 2

Figure S12: Normalized z@500-hpa qualitative results for UNet-autoregressive (S=5).

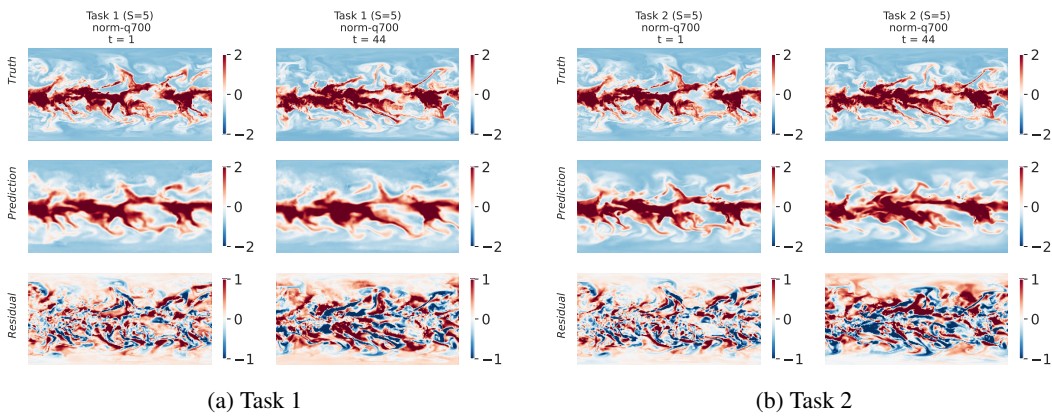

(a) Task 1          (b) Task 2

Figure S13: Normalized q@700-hpa qualitative results for UNet-autoregressive (S=5).

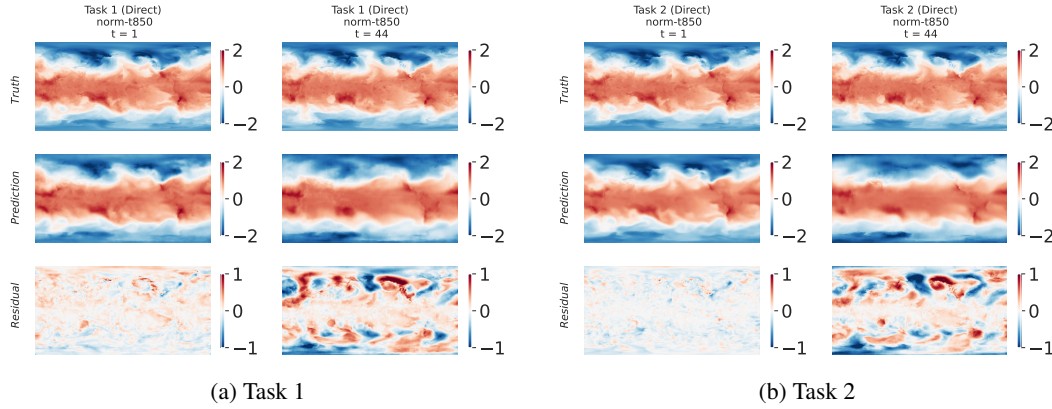

(a) Task 1          (b) Task 2

Figure S14: Normalized t@850-hpa qualitative results for UNet-direct.

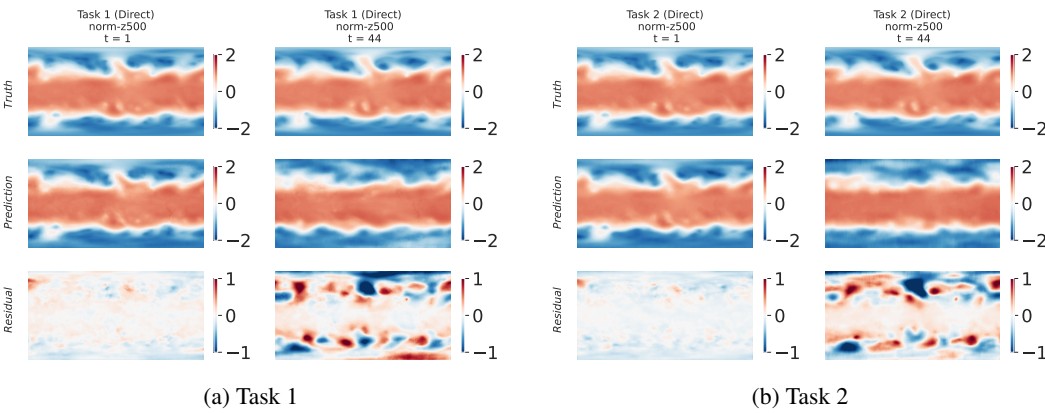

(a) Task 1          (b) Task 2

Figure S15: Normalized z@500-hpa qualitative results for UNet-direct.

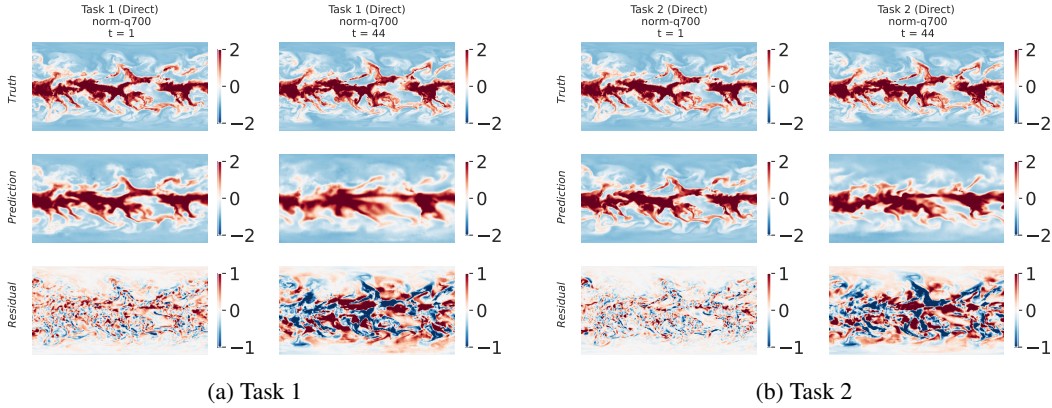

(a) Task 1          (b) Task 2

Figure S16: Normalized q@700-hpa qualitative results for UNet-direct.

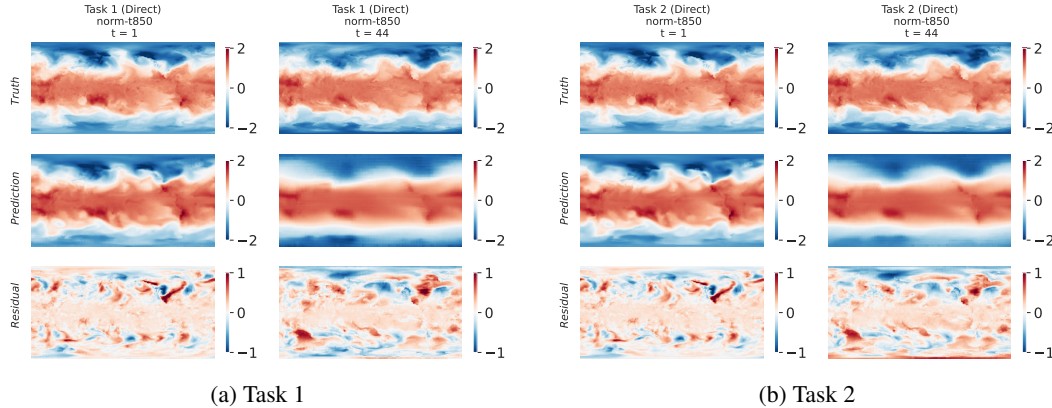

(a) Task 1               (b) Task 2

Figure S17: Normalized t@850-hpa qualitative results for ClimaX-direct.

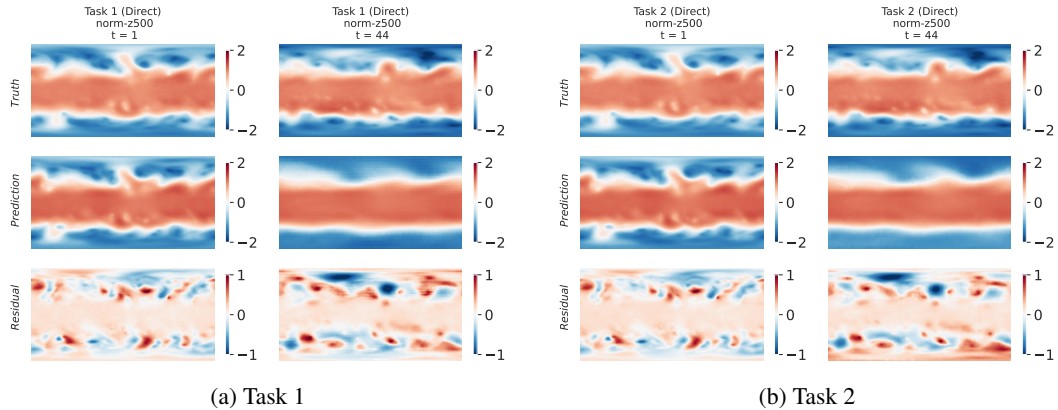

(a) Task 1               (b) Task 2

Figure S18: Normalized z@500-hpa qualitative results for ClimaX-direct.

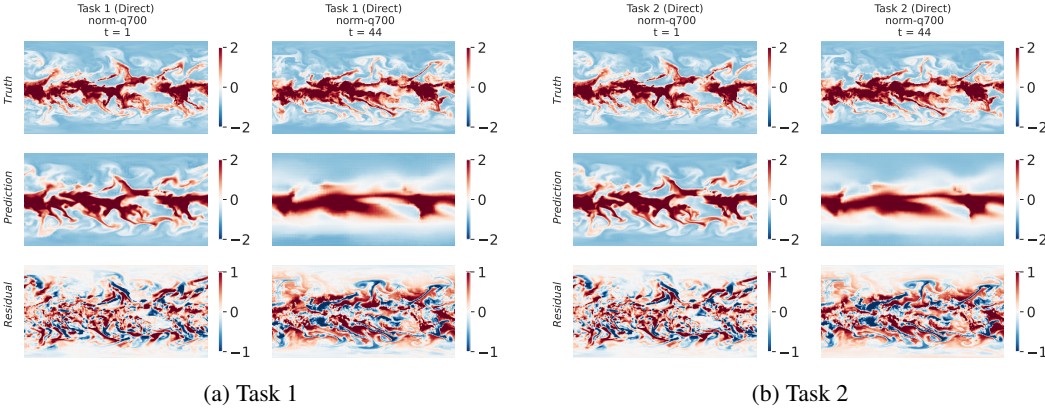

(a) Task 1               (b) Task 2

Figure S19: Normalized q@700-hpa qualitative results for ClimaX-direct.