# OpenReview forum: "ChaosBench: A Multi-Channel, Physics-Based Benchmark for Subseasonal-to-Seasonal Climate Prediction"
_NeurIPS.cc/2024/Datasets_and_Benchmarks_Track — NeurIPS 2024 Track Datasets and Benchmarks Oral_

### Official Review · Reviewer_ZL2F · 2024-07-21
**Relevant and useful benchmark for weather and climate applications**

**Rating:** 9
**Confidence:** 5
**Correctness:** Yes, the claims are correct.
**Clarity:** Yes, the paper is very well written.

**Review:**

The paper contributed ChaosBench -- a benchmark to extend the predictability range of data-driven weather/climate emulators to the Subseasonal-to-Seasonal (S2S) timescale. S2S is novel. This benchmark also includes variables beyond the typical surface-atmospheric data to include ocean, ice, and land reanalysis products for the last 45 years. It incorporates physics-based constraints to ensure explainability. The paper also evaluates the benchmark on existing AI forecast models against weather forecasts from national agencies. This work is significant to drive the existing gap in data driven weather forecasting.

**Strengths:**

The strengths of the paper include
-Novel Subseasonal-to-Seasonal benchmark for climate
-Comprehensiveness in timescale of data and variables
-Comprehensive tests of benchmarks with existing AI-based weather forecast models
-Physics constrained approach for explainability

**Additional Feedback:**

No further comments.

**Documentation:**

There are adequate documentation and information on the benchmark.

**Ethics:**

There is no ethical concern with the submission.

**Limitations:**

The authors can address how the benchmark can be adapted to other available datasets such as the MERRA-2 which has its own benefits.

**Opportunities For Improvement:**

Weather forecasts can be done at multiple resolutions. It is not clear how the benchmark will support multi-resolution forecasting approach.

**Relation To Prior Work:**

Yes.

**Summary And Contributions:**

-Novel Subseasonal-to-Seasonal benchmark for climate
-Comprehensive list of variable for climate and weather
-Rigorous evaluation against the known methods
-Physics constrained
-Exhaustive set of metrics

---

> ### Author Rebuttal · Authors · 2024-08-12
>
> We would like to thank Reviewer ZL2F for the kind feedback!
>
> > Weather forecasts can be done at multiple resolutions. It is not clear how the benchmark will support multi-resolution forecasting approach.
>
> We discuss this in the global response to reviewers.
>
> > The authors can address how the benchmark can be adapted to other available datasets such as the MERRA-2 which has its own benefits.
>
> We provide further discussion on this in the Conclusion section of our updated manuscript.
>
> _We are planning for a multi-source  reanalysis products (e.g., MERRA-2 [1]), leveraging diverse dataset strengths, such as the assimilation of different set of observations._
>
> __References__
>
> [1] Gelaro et al (2017). The modern-era retrospective analysis for research and applications, version 2 (merra-2). Journal of Climate, 30(14):5419–5454.

---

### Official Review · Reviewer_eEjR · 2024-07-25
**Important new benchmark for S2S**

**Rating:** 8
**Confidence:** 4
**Correctness:** Yes
**Clarity:** Yes

**Review:**

Subseasonal - to - Seasonal prediction remains a challenging task for physical and ML models alike. ChaosBench provides a valuable resource for driving improvements in these models, especially given the extensive land, ocean and cryosphere variables included. The comprehensive and thoughtful scoring metrics also add to the value of this benchmark and ensure against over-fitting and over-smoothing.

Pros:
 - extensive dataset including atmosphere, ocean, land and cryosphere variables over long time horizons
 - multiple physical model baselines
 - extensive and thoughtful scoring metrics
 - Good documentation

Cons:
 - None

**Strengths:**

This is an important contribution which will help drive innovation in this important task. The inclusion of ocean and land variables is especially welcome. The thoughtful selection of scoring metrics will test physical and ML models alike, and ensure the robustness of this benchmark over time.

**Additional Feedback:**

N/A

**Documentation:**

Yes, it's very good

**Limitations:**

The authors mention one limitation on spatial resolution, but temporal resolution of the ocean variables might be an issue as well. The authors state 'Since the public data is available on a monthly basis, we replicate them for daily compatibility with temporal extent between 1979 and 2023', and I'm not sure if they mean they just copy those monthly values across every day of that calendar month or something else? This seems sub-optimal since there will just be one abrupt change during the forecast period and make it difficult for the models to discern/learn the ocean evolution, potentially harming the S2S prediction.

**Opportunities For Improvement:**

It would be useful if Figure 4 and 5 were on the same y-axis to aid comparison.
It is difficult to discern the change in power at high-k in Figure 3b, would it be possible to add a visual aid to make the feature clear?

**Relation To Prior Work:**

Yes

**Summary And Contributions:**

The authors present an important new benchmark for sub seasonal to seasonal prediction, with extensive data, strong baselines and comprehensive scoring metrics. This will be a valuable resource to encourage improvements in this challenging prediction task.

---

> ### Author Rebuttal · Authors · 2024-08-12
>
> We would like to thank Reviewer eEjR for the encouraging words!
>
> > Figure 4 and 5 were on the same y-axis to aid comparison
>
> In the updated manuscript, we use the same scale for the y-axis for easier comparison while preventing plot overcrowding.
>
> > It is difficult to discern the change in power at high-k in Figure 3b, would it be possible to add a visual aid to make the feature clear?
>
> In the uploaded pdf (also referred to in Figure S8 of the updated manuscript), we provide a contrast of spectra for forecasts at t=1 and t=44, where the reported energy decay/divergence, especially at high k, is more visible.
>
> > I'm not sure if they mean they just copy those monthly values across every day of that calendar month or something else?
>
> Indeed, this is the current naive way of interpolation, and future work on advanced techniques are encouraged (as long as the available resolution is coarse). More on this in the next paragraph.
>
> >  temporal resolution of the ocean variables might be an issue as well.
>
> Although ocean procceses tend to occur at longer timescale (e.g., compared to the fast atmospheric processes), the monthly resolution of ocean variables can still be a potential source of issue. It is therefore imperative for S2S tasks to rely on probabilistic modeling (and go beyond deterministic approaches) to account for this boundary condition uncertainty. For future work, we are actively finding ways to increase the temporal resolution by e.g., stitching different products and harmonizing them for consistency, applying advanced downscaling methods, etc (each, however, comes with different pros and cons and can be a good avenue for future research). Nonetheless, since the benchmark is public, we also refer to the open-source community for recommendations.

---

### Official Review · Reviewer_d7Hd · 2024-08-03
**Well written, thorough dataset for a relevant problem.**

**Rating:** 8
**Confidence:** 3
**Clarity:** The paper is very well structured and…

**Review:**

The submission is well written and well structured:

* The prediction task is clearly defined and is highly relevant to the weather forecasting community
* The gap in the currently available datasets focused on short- or very-long-term climate is described, along with limitations of current benchmarks
* The dataset's component sources, and baseline models are fairly comprehensive, adding more NWPs than are benchmarked for the medium-range forecasting benchmarks (WeatherBench)
* The standard evaluation metrics are well understood in the weather prediction community. New metrics introduced that help evaluate the quality of weather forecasts specifically along axes where ML weather models tend to underperform. These new metrics (e.g. spectral divergence) are similar to existing metrics in WeatherBench (power spectra) but are well explored here.
* Limitations in current models are specifically described
* The code and data are very well presented, both with pseudocode and a website that provides considerable detail on data and code.

**Strengths:**

1. Clear definition of a task corresponding to a highly relevant problem that does not yet have a clear benchmark
2. High quality data and baselines, along with code and documentation to reproduce baselines
3. Well defined metrics that are relevant to the task
4. Good analysis of baseline models and their weaknesses

**Additional Feedback:**

N/A

**Correctness:**

The dataset is constructed in a sound way, along with very relevant benchmarks including both NWP and MLWP models.

**Documentation:**

There is high quality documentation available on the GitHub link shared in the paper.

**Ethics:**

No ethical concerns suspected.

**Limitations:**

There are no explicitly mentioned social limitations (nor do I see any). The authors call out a known limitation of coarse resolution data, and state future work would include higher resolution data to train and evaluate models.

**Opportunities For Improvement:**

The resolution of the data is relatively coarse, compared to both the SOTA deterministic ML weather models like GraphCast, as well as the generative modeling approaches like GenCast.

**Relation To Prior Work:**

The work discusses relevant existing benchmarks (e.g. SeasonalRodeo), and the advantages of the proposed benchmark (more variables, more baselines, larger spatiotemporal extent).

**Summary And Contributions:**

The paper proposes a dataset focused on S2S scale weather prediction, and evaluates the current set of data-driven weather prediction and NWP models on the S2S timescale. In addition to the generally well available ERA5 data, the submission also introduces other relevant data sources (ocean reanalysis and land reanalysis).

---

> ### Author Rebuttal · Authors · 2024-08-12
>
> We would like to thank Reviewer d7Hd for their constructive feedback!
>
> > The resolution of the data is relatively coarse
>
> We discuss this in the global response to reviewers.

---

### Author Rebuttal · Authors · 2024-08-12

We would like to thank the reviewers for their constructive feedback!

__On spatial resolution__

Indeed, the coarser spatial resolution of the inputs has been our point of consideration. Although the input data are available at higher (0.25-degree) resolution, the S2S forecasts from the operational physics models are not. In order for a fair comparison, the resolution of the inputs are reconciled to that of the simulation (+allowing for hybrid physics+reanalysis modeling approaches). Nevertheless, we are open-sourcing the data processing scripts with new details given in Appendix Section B.4 of the updated manuscript, allowing users to modify the resolution (highest is 0.25-degree) and affording greater flexibility to the task at hand. Formal incorporation into the benchmark is still subject to finding an appropriate mechanism that does not e.g., jeopardizes the evaluation fairness w.r.t. lower resolution physics S2S models.

_We open-source the data processing script to allow users to process the inputs given different resolution (highest is $0.25$-degree):_
```
# Process inputs with 0.25-degree resolution
$ python scripts/process_atmos.py --resolution 0.25 # ERA5
$ python scripts/process_ocean.py --resolution 0.25 # ORAS5
$ python scripts/process_land.py  --resolution 0.25 # LRA5
```

---

### Decision · Program_Chairs · 2024-09-26

**Decision:**

Accept (Oral)

**Comment:**

The reviewers see several strengths in the paper:
- s1. highly relevant problem without clear benchmark yet: subseasonal-to-seasonal climate prediction.
- s2. large, high quality data set.
- s3. inclusion of multiple domain models from physics.
- s4. clear, well defined tasks and experimental protocol.
- s5. interesting analysis of different models.
- s6. well written and documented code.

They also discussed some weaknesses:
- w1. data resolution currently relatively coarse compared to state-of-the-art
  models used for weather forecasting.
- w2. just one data resolution supported, not multiple resolutions as used in some literature.

The authors partially addressed both weaknesses in their rebuttal,
making the code to derive different resolutions available.

Overall I clearly recommend to accept the paper.